# GAQAT: Gradient-adaptive Quantization-aware Training for Domain Generalization

## Abstract

Research on loss surface geometry, such as Sharpness-Aware Minimization (SAM), shows that flatter minima improve generalization. Recent studies further reveal that flatter minima can also reduce the domain generalization (DG) gap. However, existing flatness-based DG techniques predominantly operate within a full-precision training process, which is impractical for deployment on resource-constrained edge devices that typically rely on lower bit-width representations (e.g., 4 bits, 3 bits). Consequently, low-precision quantization-aware training is critical for optimizing these techniques in real-world applications. In this paper, we observe a significant degradation in performance when applying state-of-the-art DG-SAM methods to quantized models, suggesting that current approaches fail to preserve generalizability during the low-precision training process. To address this limitation, we propose a novel Gradient-Adaptive Quantization-Aware Training (GAQAT) framework for DG. Our approach begins by identifying the scale-gradient conflict problem in low-precision quantization, where the task loss and smoothness loss induce conflicting gradients for the scaling factors of quantizers, with certain layers exhibiting opposing gradient directions. This conflict renders the optimization of quantized weights highly unstable. To mitigate this, we further introduce a mechanism to quantify gradient inconsistencies and selectively freeze the gradients of scaling factors, thereby stabilizing the training process and enhancing out-of-domain generalization. Extensive experiments validate the effectiveness of the proposed GAQAT framework. On PACS, both 3-bit and 4-bit exceed directly integrating DG and QAT by up to 4.5%. On DomainNet, our 4-bit results deliver nearly lossless performance compared to the full-precision model, while achieving improvements of up to 1.39% and 1.06% over the SOTA QAT baseline for 4-bit and 3-bit quantized models, respectively.

## 1 Introduction

Deep learning models have demonstrated remarkable performance across various computer vision tasks, such as classification (He et al., 2016; Sandler et al., 2018; Dosovitskiy, 2020), detection (Zhu et al., 2020; Zhang et al., 2022b), and semantic segmentation (Zhou et al., 2022b; Strudel et al., 2021). However, these models typically experience significant performance degradation in real-world applications due to domain shift, which manifests as poor generalization to previously unseen data distributions. Domain generalization (DG) seeks to address this challenge by enabling models trained on observed source domains to generalize effectively to unseen target domains. Strategies such as domain alignment (Li et al., 2018c; Muandet et al., 2013), data augmentation (Zhou et al., 2021; Volpi et al., 2018), and meta learning (Li et al., 2018a; Balaji et al., 2018) are commonly employed techniques. Recent studies (Gulrajani & Lopez-Paz, 2020), however, indicate that despite the development of these sophisticated techniques, basic empirical risk minimization (ERM) still yields comparable out-of-distribution generalization when experimental conditions are carefully controlled. Concurrently, growing attention has been directed towards the geometry of the loss landscape (Li & Giannakis, 2024; Foret et al., 2020; Andriushchenko & Flammarion, 2022; Wen et al., 2023) in generation, particularly the Shareness-aware Minimization (SAM) that pursues flatter minima during training. Recent works (Cha et al., 2021; Wen et al., 2023) has shown that a flatter minimum could lead to a smaller DG gap. Inspired by previous studies of flat minima (Izmailov et al., 2018; Foret et al., 2020; Liu et al., 2022; Zhuang et al., 2022; Zhang et al., 2023b; Wang et al.,

Figure 1: Illustration of GAQAT. Compared to full-precision weight gradients, the tensor-wise scale gradients have only two directions: positive and negative. For the newly introduced task-related scale gradients, we apply the GAQAT method for selective freezing. We calculate the disorder of each scale's task gradient $g_{task}$ and freeze those with disorder below a certain threshold to improve the model's generalization ability.

2023), flatness-aware methods start to gain attention and exhibit remarkable performance in domain generalization.

Despite the demonstrated effectiveness of flatness-aware methods in improving out-of-domain generalization, they are confined to *full-precision training*, which means the resulting models of current methods are not very practical to deploy. In other words, in many real-world applications, especially those involving deployment on edge devices and are truly vulnerable to domain shift environments, models operate under very computationally-constrained resources. Although the trained low-precision computations, a.k.a. the quantization-aware training (Zhou et al., 2016; Tang et al., 2022; Esser et al., 2019), have been extensively studied in I.I.D research for improving the runtime efficiency, in which the models are trained with simulated quantization during the forward-backward process and thus the weights can be aware of the numerical change, there still are challenging to achieve the generalized quantization-aware training for domain generalization, as *(a) distinct objectives:* Low precision aims to reduce model complexity, but conflicts with maintaining generalization. and *(b) training instability*: how to ensure the proper convergence for the low-precision weights as the simulated quantization and sharpness-aware minimization both involve specific gradient approximation (Wen et al., 2023; Nagel et al., 2022; Tang et al., 2024). In fact, we have observed when directly applying DG-SAM methods (Wen et al., 2023) to quantization-aware training (Esser et al., 2019; Zhou et al., 2016), there could be an unexpected degradation of the model's generalization performance (e.g., the average out-of-domain performance drops by 28.36% when quantized to 4 bits in PACS).

In this paper, we propose the Gradient-Adaptive Quantization-Aware Training (GAQAT) framework for domain generalization. Specifically, we first incorporate the smoothing factor term into the quantizer to ensure that both quantization and smoothness can be optimized jointly. Though the optimization objective seems reasonable and is optimizable, the quantizer receives two distinct gradients of the quantization and sharpness-aware minimization. By conducting a thorough analysis of the behavior of the quantizer gradients, we reveal that the significant conflicts between task loss (empirical loss) and smoothness loss induced by the gradient approximations cause the generalization ability of the trained model to degrade, even worse-performing than models optimizing a single objective. To this end, we define the *gradient disorder* that depicts the inconsistency of gradient directions during training to quantify the magnitudes of gradient conflicts. Based on this, we further design a dynamic freezing strategy, which selectively enables or disables the update of quantizers according to their gradient disorders, thus ensuring global convergence for the overall performance. The illustration of the proposed method is shown in Figure 1.

In summary, we have made the following contributions:

- We propose GAQAT, a framework to achieve efficient domain generalization by considering low-precision computations. For the first time we can empower the quantized model with good out-of-distribution generalization.

- We introduce the concept of gradient disorder to quantify gradient conflict magnitudes during optimization. Building on this, we design a dynamic freezing strategy that selectively updates quantizers based on gradient disorder, ensuring global convergence and improved generalization performance.

- Extensive experiments on PACS and DomainNet demonstrate the effectiveness of GAQAT. Specifically, on PACS, 4-bit accuracy reaches 61.33%, surpassing the baseline by 4.4%. In 3-bit, it still exceeds the baseline by 4.55%. On DomainNet, 4-bit achieves 40.74%, close to the full precision accuracy of 40.95%, while 3-bit reaches 39.53%, still outperforming the baseline.

## 2 PRELIMINARIES

### 2.1 QUANTIZATION

We consider the uniform quantization function for both weight and activation of layers: $\hat{\mathbf{v}} = Q_b(\mathbf{v}; s) = s \times \left\lfloor \text{clip}\left(\frac{\mathbf{v}}{s}, l, u\right) \right\rceil$, where $\lfloor \cdot \rceil$ denotes round-to-nearest operator, $s$ is a learnable scaling factor in QAT (Esser et al., 2019; Tang et al., 2022), and the clip function ensures values stay within the bounds $[l, u]$. In $b$-bit quantization, for activation quantization, we set $l = 0$ and $u = 2^b - 1$; for weight quantization, we set $l = -2^{b-1}$ and $u = 2^{b-1} - 1$. Furthermore, to overcome the non-differentiability of the rounding operation, the Straight-Through Estimator (STE) (Bengio et al., 2013) is employed to approximate the gradients: $\frac{\partial \mathcal{L}}{\partial \mathbf{v}} \approx \frac{\partial \mathcal{L}}{\partial \hat{\mathbf{v}}} \cdot 1_{l \leq \frac{\mathbf{v}}{s} \leq u}$.

### 2.2 FLATTER MINIMA IN DOMAIN GENERALIZATION

Following SAGM (Wang et al., 2023), we adopt three objectives for sharpness-aware minimization over the observed domains $D$: (a) empirical risk $\mathcal{L}_{ER}(\theta; D)$, (b) perturbed loss $\mathcal{L}_p(\theta; D)$, and (c) the surrogate gap $h(\theta) := \mathcal{L}_p(\theta; D) - \mathcal{L}_{ER}(\theta; D)$. Minimizing $\mathcal{L}_{ER}(\theta; D)$ and $\mathcal{L}_p(\theta; D)$ finds low-loss regions, while minimizing $h(\theta)$ ensures a flat minimum. This combination improves both training performance and generalization. Hence, the overall optimization is: $\min[\mathcal{L}_{ER}(\theta; D) + \mathcal{L}_p(\theta - \alpha \nabla \mathcal{L}_{ER}(\theta; D); D)]$ where $\alpha$ is the hyperparameter, which can be rewritten as: $\min \mathcal{L}(\theta; D) + \mathcal{L}(\theta + \hat{\epsilon} - \alpha \nabla \mathcal{L}(\theta; D); D)$ with $\hat{\epsilon} = \rho \frac{\nabla \mathcal{L}(\theta; D)}{\|\nabla \mathcal{L}(\theta; D)\|}$.

## 3 METHOD

### 3.1 QUANTIZATION IN DOMAIN GENERALIZATION

Firstly, we incorporate the smoothing factor into the quantizer to perform the generalization optimization within the latent weight space. Then, we directly employ quantization-aware training with source domains. The loss function is defined as:

$$\min \mathcal{L}_{ER}\left(Q\left(\theta; \mathbf{s}_w\right); D\right) + \mathcal{L}_p\left(Q\left(\theta - \alpha \nabla \mathcal{L}\left(Q\left(\theta; \mathbf{s}_w\right); D\right); \mathbf{s}_w\right); D\right) \tag{1}$$

However, we have observed that directly adopting this objective can lead to performance degradation, as shown in Table 2 and Table 3.

### 3.2 ANALYSIS OF THE QUANTIZER GRADIENT CONFLICT ISSUE

Compared to full-precision training, Eq. (1) has several scale factors $s_*$ in the quantizers that will correspond to two optimization targets, thus producing two sets of gradients. One set is the original task-related gradient, which we abbreviate as $\mathbf{g}_{\text{task}}$ from $\mathcal{L}_{ER}(\cdot)$, and the other is the newly introduced flatness-related gradient, abbreviated as $\mathbf{g}_{\text{smooth}}$ from $\mathcal{L}_p(\cdot)$.

However, the scale factor, used to portray the characteristic of weight and activation distribution Tang et al. (2022), is highly sensitive to the perturbations Esser et al. (2019); Liu et al. (2023).

Table 1: Performance results for perturbed scaling factors in the 4-bit test on Clipart and Infograph datasets from DomainNet. The notation x% indicates a scaling factor change by x%. Red highlights performance degradation, while green signifies improvement. These results suggest that the apparent convergence of scaling factors towards a suboptimal state does not necessarily imply satisfactory convergence and can negatively affect OOD performance.

| Layer | origin | 80% | 90% | 110% | 120% |
|---|---|---|---|---|---|
| layer3.0.conv1.w.s | 60.21 / 15.81 | 60.30 / 15.93 | 60.15 / 15.94 | 59.96 / 15.62 | 59.82 / 15.38 |
| layer3.0.conv1.a.s | 60.21 / 15.81 | 60.47 / 16.12 | 60.31 / 15.90 | 60.10 / 15.72 | 59.93 / 15.65 |
| layer1.0.conv1.w.s | 60.21 / 15.81 | 60.25 / 15.60 | 60.14 / 15.61 | 60.32 / 15.48 | 60.18 / 15.27 |
| layer1.0.conv1.a.s | 60.21 / 15.81 | 60.23 / 15.81 | 60.22 / 15.85 | 60.26 / 15.78 | 60.24 / 15.67 |

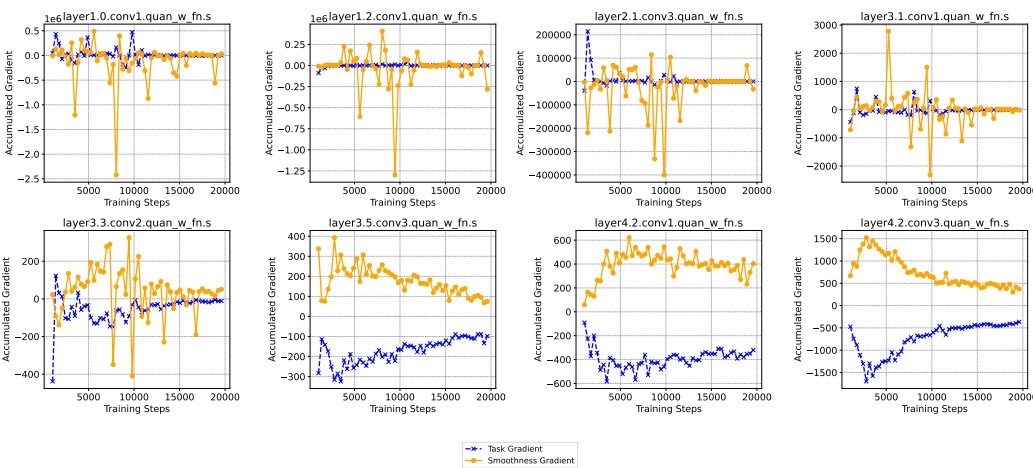

Figure 2: Results of cumulative gradients every 350 steps in the 4-bit test on the PACS ART domain, revealing conflicts in the scaling factors.

We therefore have the following hypothesis for the scaling factor in quantizer: *The apparent convergence of scaling factors reaching a sub-optimal state does not necessarily indicate satisfactory convergence and can negatively impact OOD performance.* To verify this hypothesis, we perform perturbations on the scales of certain layers in the trained model by further scaling them by $x \in \{0.8, 0.9, 1.1, 1.2\}$ times. As shown in Table 1, perturbing the scale to certain layers significantly improves OOD performance, while in other layers, it results in performance degradation. This indicates the proper convergence of quantization parameters (the scaling factor in the quantizer) is of importance for out-of-distribution generalization, proving that the scale converges suboptimally due to the conflicted gradients of two objectives. To further show the interference between $\mathbf{g}_{smooth}$ and $\mathbf{g}_{task}$, we visualized the sum of these two gradients during the training process. As shown at the top of Figure 2, a significant gradient conflict is evident. Morever, for certain layers, $\mathbf{g}_{task}$ and $\mathbf{g}_{smooth}$ is opposite and tend to cancel each other out (bottom of Figure 2). This suggests that the scaling factors of these layers are approaching a state we define as the sub-optimal equilibrium state. Since both simulated quantization and sharpness-aware minimization involve specific gradient approximations and according to (Liu et al., 2023), the weight oscillations caused by the discrete nature of quantization can be significantly amplified by learnable scaling factors, the conflict between $\mathbf{g}_{task}$ and $\mathbf{g}_{smooth}$ can substantially negatively impact the performance of QAT in DG scenarios.

## 3.3 SELECTIVE FREEZING TO RESOLVE GRADIENT CONFLICTS

To address the issue of scaling factor gradient conflicts, we propose Gradient-Adaptive Quantization-Aware Training (GAQAT) framework for domain generalization, a selective freezing training strategy. First, we define the *gradient disorder* to quantify the inconsistency of gradient directions during training.

**Definition 3.1. Gradient Disorder:** Suppose we have $K$ steps of training, and at each step $j$, this step's gradient is formalized as $\mathbf{g}_j$. We define two gradient sequences: $S_1 = \{\mathbf{g}_1, \mathbf{g}_2, \ldots, \mathbf{g}_{K-1}\}$ and

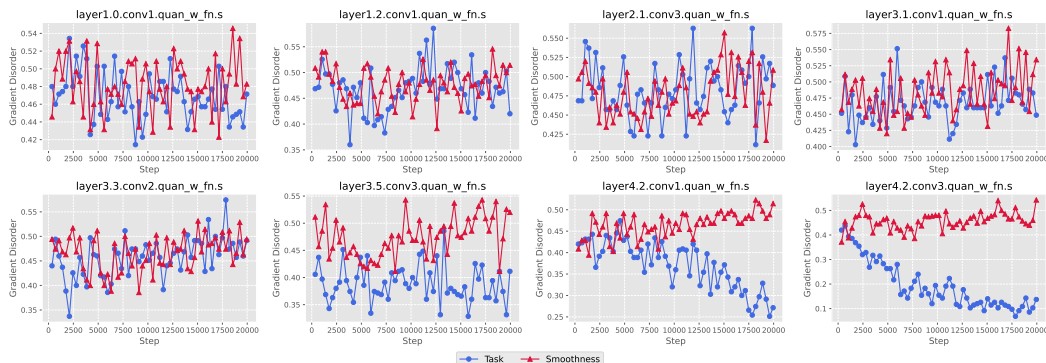

Figure 3: Results of task and smoothness gradient disorder of scaling factors over 350 steps in the 4-bit test on the PACS ART domain, revealing in some layers, the gradient disorder of the $\mathbf{g}_{\text{task}}$ decreases significantly as training progresses.

$S_2 = \{\mathbf{g}_2, \mathbf{g}_3, \ldots, \mathbf{g}_K\}$. Let $\text{sgn}(\cdot)$ denote the element-wise sign function. The gradient disorder is defined as:

$$\delta = \frac{1}{K} \mathbb{1} \left( \text{sgn}(\mathbf{S_1}) \neq \text{sgn}(\mathbf{S_2}) \right), \tag{2}$$

where $\mathbb{1}(\cdot)$ is the indicator function. $\delta$ represents the proportion of steps where the gradient direction is opposite to that of the previous step.

A lower gradient disorder indicates more consistent gradient directions, which implies more stable training. It is important to note that while a high disorder does not necessarily indicate incorrect gradients, a low disorder can provide some assurance of gradient correctness.

Figure 3 indicates that in some layers, the gradient disorder of the $\mathbf{g}_{\text{task}}$ decreases significantly as training progresses. This suggests that the gradient direction of the $\mathbf{g}_{\text{task}}$ becomes increasingly consistent, which is somewhat counterintuitive. In contrast, the gradient disorder of the flatness scaling factor shows no significant change across layers. And layers with lower task gradient disorder (as shown in the three images at the bottom-right in Figure 3) exhibit a clear phenomenon of opposite and similarmagnitude gradients in Figure 2. This indicates that layers with lower task gradient disorder are more likely to settle into sub-optimal equilibrium state.

**Algorithm 1** Dynamic Selective Freezing Strategy for Scaling Factors

**Require:** Training steps $T$, evaluation interval $K$, disorder threshold $r$, set of scaling factors $\{S_1, S_2, \ldots, S_n\}$
1: Initialize step counter $t \leftarrow 0$, freeze$[S_i] \leftarrow$ False for all $S_i$
2: **while** $t < T$ **do**
3:     **for** each scaling factor $S_i$ **do**
4:         **if** freeze$[S_i] =$ True **then**
5:             Update $S_i$ using only $\mathbf{g}_{\text{smooth}}$
6:         **else**
7:             Update $S_i$ using both $\mathbf{g}_{\text{task}}$ and $\mathbf{g}_{\text{smooth}}$
8:         **end if**
9:     **end for**
10:     **if** $t \bmod K = 0$ **then**
11:         **for** each scaling factor $S_i$ **do**
12:             Compute gradient disorder $\delta_{t,S_i}$
13:             **if** $\delta_{t,S_i} < r$ **then**
14:                 freeze$[S_i] \leftarrow$ True
15:             **else**
16:                 freeze$[S_i] \leftarrow$ False
17:             **end if**
18:         **end for**
19:     **end if**
20:     $t \leftarrow t + 1$
21: **end while**

These observations suggest that the training of the $\mathbf{g}_{\text{task}}$ gradients may interfere with the training of the flatness scaling factor. Inspired by the gradient freezing strategies(Liu et al., 2023; Tang et al., 2024; Nagel et al., 2022), we propose discarding $\mathbf{g}_{\text{task}}$ in certain scales to mitigate these conflicts.

**Assumption 3.1. Impact of Incomplete Scaling Factor Convergence to other layers:** The apparent convergence of scaling factors reaching a suboptimal equilibrium state between task and flatness objectives could impact other layers, including causing outlier gradients.

To verify this hypothesis, we conducted an experiment using the gradient disorder of $\mathbf{g}_{\text{task}}$ as an indicator of convergence (see Figure 4). The results demonstrate that the frozen scaling factor continues

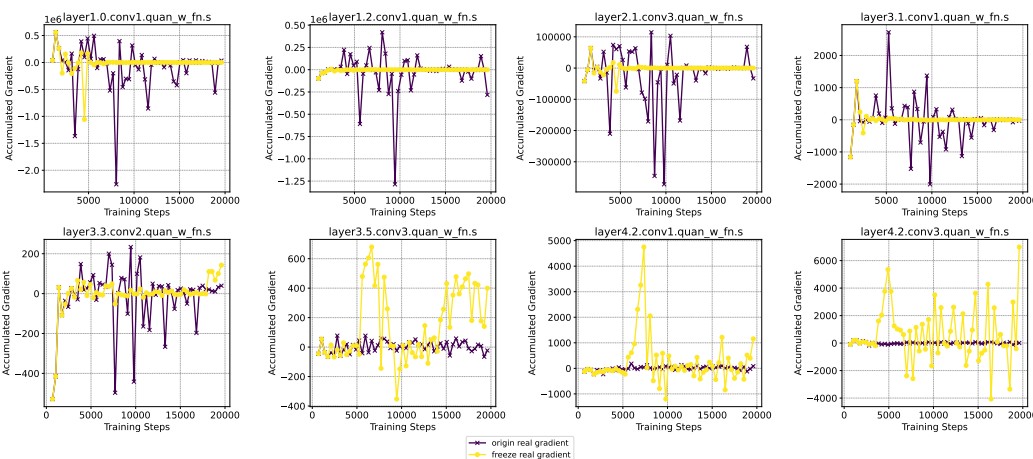

Figure 4: Results of freezing over 350 steps in the 4-bit test on the PACS ART domain, using gradient disorder as an indicator, with no unfreezing. The findings suggest that instability in gradient fluctuations is partly caused by interference between scaling factors during training. Moreover, the gradient disorder indicator proves to be a useful metric for determining when to freeze.

to be updated via $\mathbf{g}_{\text{smooth}}$, and the gradient fluctuations in unfrozen layers are significantly reduced. This suggests that the instability in gradient fluctuations is partly caused by interference between scaling factors during training.

Based on these findings, we propose a selective freezing strategy to address scaling factor instability and improve flatness convergence. Persistently freezing the $\mathbf{g}_{\text{task}}$ of certain layers without selectively unfreezing them in specific cases may result in suboptimal convergence. Therefore, we adopt a dynamic approach. Every $K$ steps, we evaluate the gradient disorder. If the disorder $\delta_{t,S_i}$ for scaling factor $S_i$ at step $t$ is below a threshold $r$, we freeze the $\mathbf{g}_{\text{task}}$ of $S_i$ for the next $K$ steps; otherwise, we continue updating it. This dynamic selective freezing strategy allows the flatness of scaling factor to continue training while mitigating the adverse effects of gradient conflicts. By periodically reassessing and adjusting which scaling factors are frozen, we aim to improve overall convergence and enhance the model's generalization performance in DG scenarios. Full process is summarized in Algorithm 1.

## 4 EXPERIMENT

### 4.1 EXPERIMENTAL SETUP AND IMPLEMENTATION DETAILS

**Quantization.** We follow established practices in Quantization-Aware Training (QAT) literature by employing the LSQ-type method (Esser et al., 2019) to quantize both weights and activations. The quantization scaling factors are learned with a fixed learning rate of $1 \times 10^{-5}$. We use Mean Squared Error (MSE) range estimation (Nagel et al., 2021) to determine the quantization parameters for weights and activations. Due to the risk of test data information leakage of supervised pre-trained weights revealed by Yu et al. (2024b), we employ MoCo-v2 (Chen et al., 2020) pretrained ResNet-50 as initialization as recommended. Then we fine-tune the model using Empirical Risk Minimization (ERM) to obtain a full-precision model with generalization capabilities, which serves as the baseline for quantization. The weights and activations are fully quantized, except for the first convolutional layer, which quantizes only the activations, and the final linear layer, which remains unquantized, striking a balance between efficiency and model capacity. We evaluate the performance under extremely low bit-width conditions of 3 and 4 bits.

**Datasets and evaluation protocol.** We conduct a comprehensive evaluation on two widely used DG datasets: PACS (Li et al., 2017), containing 9,991 images across 7 categories and 4 domains, and DomainNet (Peng et al., 2019), consisting of 586,575 images across 345 categories and 6 domains. We basically follow the evaluation protocol of DomainBed (Gulrajani & Lopez-Paz, 2020),

including the optimizer, data split, and model selection, where we adopt test-domain validation as our model selection strategy for all algorithms in our experiments. For PACS, for each time we treat one domain as the test domain and other domains as training domains, which is the leave-one-domain-out protocol commonly adopted in DG. For DomainNet, following Yu et al. (2024b), we divide the domains into three groups: (1) *Clipart* and *Infograph*, (2) *Painting* and *Quickdraw*, and (3) *Real* and *Sketch*. Then we employ the leave-one-group-out protocol, where we treat one group of two domains as test domains and other two groups as training domains each time. For the number of training steps, for full-precision models we set it as 5,000 for PACS and 15,000 for DomainNet following Cha et al. (2021), while for quantization training we use 20,000 for PACS and 50,000 for DomainNet. To reduce time cost, for quantization training we conduct validation and testing for DomainNet only after 45,000 steps.

**Hyperparameter settings.** Given the substantial computational resources required by the original DomainBed setup, we adjust the hyperparameter search space and conduct grid search to reduce computational cost following SAGM (Wang et al., 2023). The search space of learning rate is {1e-5, 3e-5, 5e-5}, and the dropout rate is fixed as zero. The batch size of each training domain is set as 32 for PACS and 24 for DomainNet. Following SAM (Foret et al., 2020), we fix the hyperparameter $\rho = 0.05$. Following SAGM (Wang et al., 2023), we set $\alpha$ in Equation (1) as 0.001 for PACS and 0.0005 for DomainNet, and set weight decay as 1e-4 for PACS and 1e-6 for DomainNet.

For PACS, the gradient disorder threshold $r$ is selected from {0.28, 0.30, 0.32} for both 3-bit and 4-bit quantization. The number of freeze steps is selected from {300, 350, 400} for 4-bit quantization, and from {100, 150, 200} for 3-bit quantization. For DomainNet, $r$ is selected from {0.20, 0.25} for 4-bit quantization, and from {0.02, 0.03} for 3-bit quantization. The number of freeze steps is chosen from {3000, 4000} for 4-bit quantization, and from {200, 300} for 3-bit quantization, as we observed that conflicts are more severe in 4-bit than in 3-bit quantization. To reduce the high computational cost, we first select the shared hyperparameters, i.e. learning rate, weight decay, through grid search, which serve as the base hyperparameter configuration. Then we fix the base configuration and conduct further grid search on our specific hyperparameters, i.e. freeze steps, freeze threshold.

## 4.2 MAIN RESULTS

We evaluated our method on the PACS and DomainNet datasets, comparing it to existing approaches (see Tables 2 and 3). Our method achieves the best performance across different quantization bit-widths (4/4 and 3/3). At 4-bit quantization, it attains an average test accuracy of **61.33%** on PACS, outperforming LSQ (**58.98%**) and SAGM+LSQ (**56.93%**); When the quantization bit-width is reduced to 3 bits, our method maintains superior performance with an average accuracy of **57.13%**, remain the best, demonstrating its robustness.

Table 2: Results on PACS dataset.

| Method | Bit-width (W/A) | Art (val/test) | Cartoon (val/test) | Photo (val/test) | Sketch (val/test) | Avg (val/test) |
|---|---|---|---|---|---|---|
| ERM | Full | 96.63/84.62 | 95.79/80.86 | 96.78/95.73 | 96.48/79.96 | 96.42/85.29 |
| LSQ | 4/4 | **88.28/51.07** | **78.74**/58.10 | 80.81/63.77 | 74.96/62.98 | 80.70/58.98 |
| SAGM+LSQ | 4/4 | 86.21/46.49 | 76.86/55.12 | 81.79/64.67 | 70.60/61.45 | 78.87/56.93 |
| Ours | 4/4 | 86.75/49.24 | 78.11/**59.22** | **85.31/69.46** | **77.25/67.40** | **81.86/61.33** |
| LSQ | 3/3 | 82.07/39.29 | 74.97/**58.69** | **79.21**/59.28 | 74.88/**64.41** | 77.78/55.42 |
| SAGM+LSQ | 3/3 | 83.48/43.56 | 72.34/52.45 | 74.22/58.16 | 64.94/56.14 | 73.75/52.58 |
| Ours | 3/3 | **84.43/44.36** | **75.77**/59.06 | 76.85/**61.75** | **75.70**/63.33 | **78.19/57.13** |

On the DomainNet dataset, at 4-bit quantization, our method achieves an average test accuracy of **40.74%**, surpassing both LSQ and SAGM+LSQ, and nearing the full-precision accuracy of **40.95%**, consistently delivering the best performance across all domains. With 3-bit quantization, it achieves **39.53%**, maintaining the best performance, though with a slight drop in validation accuracy. We observed fewer scale gradient conflicts in 3-bit compared to 4-bit (see Figure 5), where task gradients dominate. This explains the slight validation drop when freezing task gradients, supporting the effectiveness of our approach.

Table 3: Results on DomainNet dataset.

| Method | Bit-width (W/A) | Clipart | Infograph | Painting | Quickdraw | Real | Sketch | Avg |
|---|---|---|---|---|---|---|---|---|
| ERM | Full | 66.80/59.42 | 66.80/18.30 | 61.13/47.90 | 61.13/13.78 | 58.20/57.82 | 58.20/48.46 | 62.04/40.95 |
| LSQ | 4/4 | 66.34/60.45 | 66.34/15.65 | 59.56/44.69 | 59.56/14.76 | 57.82/52.70 | 57.82/47.82 | 61.24/39.35 |
| SAGM+LSQ | 4/4 | 65.77/60.73 | 65.77/15.64 | 61.21/46.67 | 61.21/16.29 | 56.77/52.22 | 56.77/48.45 | 61.25/40.00 |
| Ours | 4/4 | **67.20**/**61.00** | **67.20**/**16.12** | **62.51**/**47.80** | **62.51**/**16.44** | **58.59**/**53.45** | **58.59**/**49.63** | **62.77**/**40.74** |
| LSQ | 3/3 | 62.90/58.28 | 62.90/14.16 | **58.84**/**43.90** | **58.84**/14.53 | 57.48/52.36 | 57.48/47.56 | 59.74/38.47 |
| SAGM+LSQ | 3/3 | 63.00/**58.55** | 63.00/**15.01** | 57.61/43.22 | 57.61/16.39 | **59.23**/53.73 | **59.23**/49.74 | **59.95**/39.44 |
| Ours | 3/3 | **63.07**/58.50 | **63.07**/14.97 | 57.69/43.35 | 57.69/**16.40** | 58.68/**54.01** | 58.68/**49.97** | 59.81/**39.53** |

## 4.3 ABLATION STUDY

In our analysis, we validated the effectiveness of freezing $g_{task}$ with gradient disorder below a specific threshold and periodically reselecting the freeze set to stabilize quantization training in the DG scenario. A natural question arises: what if we reverse these choices? Specifically, what happens if we freeze scaling factors with gradient disorder above the threshold, or if we do not unfreeze after freezing?

As shown in Table 4, we fixed the freeze steps at 350 and set the threshold at 0.3 on the PACS dataset. We denote the strategy of freezing scaling factors above the threshold (with reselection) as *Ours (Reverse Ratio)* and continuous freezing without unfreezing as *Ours (w/o Unfreeze)*. It can be seen that simply not unfreezing still leads to a certain improvement in OOD performance. However, if we apply reverse freezing, it significantly decreases performance on both the validation and test sets. This further validating the effectiveness of our proposed method.

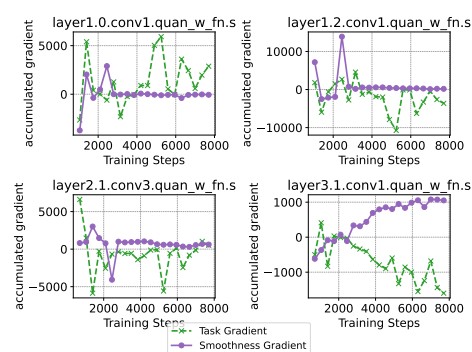

Figure 5: Results of cumulative gradients every 2111 steps in the 3-bit test on the DoaminNet Clipart and Infograph domains, revealing fewer anomalous gradients compared to 4-bit, with $g_{task}$ dominating.

Table 4: Ablation Study on PACS: Effect of Freezing Strategies

| Method | Bit-width (W/A) | Art (val/test) | Cartoon (val/test) | Photo (val/test) | Sketch (val/test) | Avg (val/test) |
|---|---|---|---|---|---|---|
| SAGM+LSQ | 4/4 | 86.21/46.49 | 76.86/55.12 | 81.79/64.67 | 73.61/58.81 | 79.62/56.27 |
| Ours (Reverse Ratio) | 4/4 | 84.64/45.21 | 78.01/55.33 | 77.61/60.10 | 74.33/59.80 | 78.65/55.11 |
| Ours (w/o Unfreeze) | 4/4 | 86.81/**48.51** | 77.87/56.66 | 77.61/60.10 | 75.71/**63.55** | 79.5/57.21 |
| Ours | 4/4 | **87.45**/48.20 | **78.11**/**59.22** | **83.48**/**67.51** | **75.75**/62.37 | **81.20**/**59.33** |

Additionally, we analyzed the sensitivity of different domains to hyperparameter settings using the 4-bit configuration on PACS. We fixed the number of freeze steps and varied the threshold, as shown in Tables 5 and 6. The results indicate that different domains exhibit varying sensitivities to hyperparameters. Within a certain reasonable range, it is the level of gradient disorder threshold that ultimately determines performance, while the step size remains relatively insensitive. Therefore, establishing distinct hyperparameter search spaces for each domain could lead to improved performance.

## 4.4 LOSS SURFACE VISUALIZATION

Following the approach in (Li et al., 2018b), Figure 6 illustrates the differences in loss surface visualizations across the four domains of PACS when incorporating SAGM directly versus applying our proposed method. The results clearly show that our method consistently achieves significantly smoother loss surfaces across all four domains.

Table 5: Ablation Study on PACS: Effect of Freeze Steps

| Freeze Steps | Bit-width (W/A) | Art (val/test) | Cartoon (val/test) | Photo (val/test) | Sketch (val/test) | Avg (val/test) |
|---|---|---|---|---|---|---|
| 300 | 4/4 | 85.85/47.65 | 78.05/58.37 | **84.32**/69.09 | 74.35/61.80 | 80.64/59.23 |
| 350 | 4/4 | **87.45**/48.20 | 78.11/**59.22** | 83.48/67.51 | 75.75/62.37 | **81.20/59.33** |
| 400 | 4/4 | 86.51/**48.38** | **78.44**/56.66 | 82.77/**69.09** | **76.67/62.53** | 81.10/59.17 |

Table 6: Ablation Study on PACS: Effect of Threshold $r$

| Threshold $r$ | Bit-width (W/A) | Art (val/test) | Cartoon (val/test) | Photo (val/test) | Sketch (val/test) | Avg (val/test) |
|---|---|---|---|---|---|---|
| 0.28 | 4/4 | 86.74/**49.24** | 77.79/55.92 | 79.77/64.22 | 74.59/62.21 | 79.72/57.90 |
| 0.30 | 4/4 | **87.45**/48.20 | **77.87/56.45** | 79.46/63.62 | 75.75/62.37 | 80.13/57.66 |
| 0.32 | 4/4 | 86.96/48.63 | 77.20/55.17 | **80.62/64.60** | **77.25/67.40** | **80.51/58.95** |

# 5 RELATED WORK

## 5.1 DOMAIN GENERALIZATION

In practical applications, when deploying machine learning models, test data distribution may differ from the training distribution, a common phenomenon known as distribution shift (Liu et al., 2021; Yu et al., 2024a; Koh et al., 2021). Domain generalization (DG) aims to enhance a model's ability to generalize to unseen domains (Wang et al., 2022; Zhou et al., 2022a). Common strategies include domain alignment (Muandet et al., 2013; Li et al., 2018c; Zhao et al., 2020), meta learning (Li et al., 2018a; Balaji et al., 2018; Dou et al., 2019), data augmentation (Zhou et al., 2021; Carlucci et al., 2019), disentangled representation learning (Zhang et al., 2022a) and utilization of causal relations (Mahajan et al., 2021; Lv et al., 2022). Inspired by previous studies of flat minima (Izmailov et al., 2018; Foret et al., 2020; Liu et al., 2022; Zhuang et al., 2022; Zhang et al., 2023b), flatness-aware methods start to gain attention and exhibit remarkable performance in domain generalization (Cha et al., 2021; Wang et al., 2023; Zhang et al., 2023a), such as SAGM(Wang et al., 2023), which improves generalization by optimizing the angle between weight gradients. However, these methods primarily focus on full-precision models, which are impractical for deployment on edge devices commonly used in high-risk scenarios and do not take into account the factors specific to quantization. We specifically focus on strategies to enhance model generalization in quantized training environments.

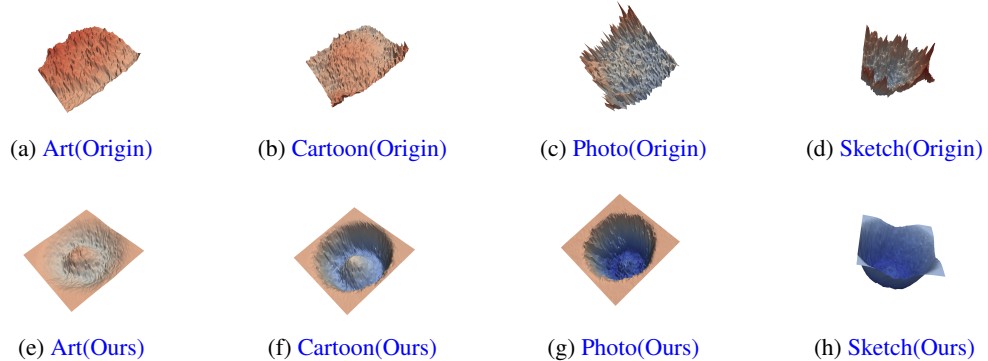

| (a) Art(Origin) | (b) Cartoon(Origin) | (c) Photo(Origin) | (d) Sketch(Origin) |
|---|---|---|---|
| (e) Art(Ours) | (f) Cartoon(Ours) | (g) Photo(Ours) | (h) Sketch(Ours) |

Figure 6: Visualization of the loss landscape across various domains. Top is the direct integration of SAGM into QAT, bottom is proposed method. Our method achieves smoother loss surfaces across all four domains in PACS.

## 5.2 QUANTIZAION-AWARE TRAINING

Quantizaion-aware training (QAT) involves inserting simulated quantization nodes and retraining the model, which achieves a better balance between accuracy and compression ratio (Hubara et al., 2021; Nagel et al., 2020). DoReFa (Zhou et al., 2016) and PACT (Choi et al., 2018) use low-precision weights and activations during the forward pass and utilize STE techniques (Bengio et al., 2013) during backpropagation to estimate gradients of the piece-wise quantization functions. LSQ (Esser et al., 2019) adjusts the quantization function by introducing learnable step size scaling factors. Recently, some works have explored the possibility of improving quantization performance by freezing unstable weights to further enhance results (Nagel et al., 2022; Tang et al., 2024; Liu et al., 2023); however, these methods have only considered the Identically Distributed (I.I.D.) assumptions. Due to distribution shifts in unseen data—which often occur in practical applications—the quality and reliability of quantized models cannot be guaranteed (Hu et al., 2022).

## 6 CONCLUSION AND FUTURE WORK

In this paper, we propose GAQAT for domain generalization. We introduce a smoothing factor into the quantizer to jointly optimize quantization and smoothness. Our analysis of quantizer gradients revealed significant conflicts between task loss and smoothness loss due to gradient approximations, impacting generalization. To address this, we define gradient disorder to quantify quantizer gradient conflicts and designed a dynamic freezing strategy that selectively updates quantizers based on disorder levels, ensuring global performance convergence. Extensive experiments on PACS and DomainNet, along with ablation studies, demonstrate the effectiveness of GAQAT.

**Limitations and future work.** Although we incorporated SAGM's smoothing objective into quantization, other smoothing objectives may also impact scaling factor gradients, suggesting future research potential. Our experiments reveal varying domain sensitivity to scaling factor gradients, but we only examined conflicts between task and flatness objectives. The relationship between domains and scaling factors remains unexplored.

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
