# OpenReview forum: "GAQAT: Gradient-Adaptive Quantization-Aware Training for Domain Generalization"
_ICLR.cc/2025/Conference — Submitted to ICLR 2025_

### Official Review · Reviewer_TWZa · 2024-10-27

**Soundness:** 2
**Presentation:** 2
**Contribution:** 2
**Rating:** 3
**Confidence:** 4

**Summary:**

This paper combines quantization-aware training with the domain generalization model. At first, the authors identify the scale factors (used in quantization) suffer from a gradient conflict problem, where the task loss and smoothness loss present conflicting gradients. As a result, GAQAT, which dynamically quantifies gradient conflicting and selectively freezes the gradients of scaling factors, is proposed to stabilize the training process and enhance domain generalization performance.

**Strengths:**

1 This paper provides detailed implementation settings to help re-production.
2 The experimental results are comprehensive, which helps prove the effectiveness of GAQAT.

**Weaknesses:**

I have some concerns about the writing and experiment comparison of this paper.

Writing:

1 In the main text, Figure 3 should be referred before Figure 4.

2 The gradient disorder is measured with two sequences according to Equation 4. However, in Line 254, the author says 'the gradient disorder of the  g_{task} decreases significantly as training progresses.' How do measure the gradient disorder of the g_{task} alone?

3 Line 259 - 263. Why lower layers with lower gradient disorder are more likely to exhibit the phenomenon of two gradients in suboptimal equilibrium state? How the Figure 2 and Figure 4 prove this? What is the suboptimal equilibrium state?

4 Totally lost when reading Line 287-294

5 Line 297, 'Continuously **freezing** the g_{task} without **unfreezing**'? How can you freeze g_{task} while unfreezing it?


Experiment comparison:

1 Why only 4/4 and 3/3 bit-width?
2 Maybe the author should add some experiments that quantized the trained model with PTQ methods.
3 Does the quantized model is initialized from the pre-trained full-precision model?

**Questions:**

See Weaknesses

---

> ### Author Response · Authors · 2024-11-23
> **Response to Reviewer TWZa**
>
> **W1**
>
> > In the main text, Figure 3 should be referred before Figure 4.
>
> Thank you for your suggestion, it has been updated in the latest version.
>
> **W2**
>
> > The gradient disorder is measured with two sequences according to Equation 4. However, in Line 254, the author says 'the gradient disorder of the g_{task} decreases significantly as training progresses.' How do measure the gradient disorder of the g_{task} alone?
>
> Gradient disorder measures the inconsistency in directions between adjacent gradients within the same gradient sequence (applicable to any gradient sequence, see lines 211-235). In our experiments, it is computed every KK steps (see lines 294-295). For example, given $K=3 \ G={−1,2,3,−4} S_1 = \{-1, 2, 3\}, \quad S_2 = \{2, 3, -4\},\text{sgn}(S_1) = \{-1, 1, 1\}, \quad \text{sgn}(S_2) = \{1, 1, -1\},\text{sgn}(S_1) \neq \text{sgn}(S_2) = 2.Thus, the gradient disorder is:\delta = \frac{1}{3} \cdot 2 = \frac{2}{3}.$
>
> **W3**
>
> > Line 259 - 263. Why lower layers with lower gradient disorder are more likely to exhibit the phenomenon of two gradients in suboptimal equilibrium state? How the Figure 2 and Figure 4 prove this? What is the suboptimal equilibrium state?
>
> Gradients with similar magnitudes but opposite directions define what we term as the "opposite equilibrium state." We have clarified this in the latest version and reordered the figures for better comprehension. Specifically, layers with lower gradient disorder (illustrated in the bottom-right images of Figure 3) show this phenomenon clearly in Figure 2.
>
> **W4**
>
> > Line 259 - 263. Why lower layers with lower gradient disorder are more likely to exhibit the phenomenon of two gradients in suboptimal equilibrium state? How the Figure 2 and Figure 4 prove this? What is the suboptimal equilibrium state?
>
> **It demonstrates that sub-optimal equilibrium state leads to suboptimal convergence in other layers, causing abnormal gradients.** To address this, we froze layers with task gradient disorder below a certain threshold. Figure 4 shows this reduced abnormal gradients in the unfrozen layers, while the gradients in the frozen layers fluctuated and were further refined through continued training.
>
> **W5**
>
> > Line 297, 'Continuously **freezing** the g_{task} without **unfreezing**'? How can you freeze g_{task} while unfreezing it?
>
> "Unfreezing" refers to the process where previously frozen layers are re-activated under certain conditions, allowing **g_task** to rejoin training. "Not unfreezing" means the frozen layers remain inactive, with **g_task** excluded from training once frozen.
>
> **E1**
>
> > Why only 4/4 and 3/3 bit-width?
>
> For 5-bit and 8-bit quantization, we observed similar gradient conflict phenomena. Results on the *cartoon* domain show improvements even without fine-tuned hyperparameter search: 5-bit accuracy increased from 76.05%/54.64% to 80.59%/59.22%, and 8-bit from 77.42%/56.34% to 80.74%/56.45%. Due to time and resource constraints, we will provide complete results with more refined hyperparameter search in the final version. The newly added Figure 6 visualizes the loss curves for the four PACS domains under 4-bit quantization. As shown, the proposed method achieves significantly smoother surfaces, further confirming its effectiveness.
>
> Additionally, a classic study on freezing also explored 3-bit and 4-bit quantization settings [1]
>
> [1] Nagel M, Fournarakis M, Bondarenko Y, et al. Overcoming oscillations in quantization-aware training[C]//International Conference on Machine Learning. PMLR, 2022: 16318-16330.
>
>
>
> **E2**
>
> >  Maybe the author should add some experiments that quantized the trained model with PTQ methods.
>
> QAT has shown strong performance in non-large-scale models, which is why we focused on it. We believe that PTQ is a promising future direction, and PTQ could be explored in future work, though it is not the focus of this paper.
>
> **E3**
>
> >  Does the quantized model is initialized from the pre-trained full-precision model?
>
> Our experiments are based on a model pre-trained on the DG source domain dataset (lines 309-315). Specifically, **we used the MoCoV2-pretrained ResNet50, as proposed in [1] , and fine-tuned it using ERM on the source domain data from the DG dataset.** This approach aligns with current standard DG practices, where a pretrained model is fine-tuned on a DG dataset for better performance, as outlined in previous works like [1] [2] [3]. Based on this well-performing full-precision DG model, we proceeded with quantization.
>
> [1] Yu, Han, et al. "Rethinking the evaluation protocol of domain generalization." *Proceedings of the IEEE/CVF Conference on Computer Vision and Pattern Recognition*. 2024.
>
> [2] Gulrajani, Ishaan, and David Lopez-Paz. "In search of lost domain generalization." *arXiv preprint arXiv:2007.01434* (2020).
>
> [3] Cha, Junbum, et al. "Swad: Domain generalization by seeking flat minima." *Advances in Neural Information Processing Systems* 34 (2021): 22405-22418.

---

> > ### Comment · Reviewer_TWZa · 2024-11-26
> >
> > Dear author,
> >
> > I highly appreciate this rebuttal. However, I still consider this rebuttal to confuse me.
> >
> >
> > 1 According to the rebuttal of W2, the Gradient disorder is measured by using inconsistency in directions between gradients sequence.
> >
> > However, in the rebuttal of W3, the author explains that lower layers with lower gradient disorder are more likely to exhibit the phenomenon of two gradients in suboptimal equilibrium state. Meanwhile, suboptimal equilibrium means similar magnitudes but opposite directions (Gradients with similar magnitudes but opposite directions define what we term the "opposite equilibrium state").
> >
> > This really confused me. If a layer has a lower gradient disorder, it should have **low inconsistency in directions** according to the rebuttal of W2. But in the rebuttal of W3, lower layers with lower gradient disorder are more likely to exhibit suboptimal equilibrium state, which means it should have **high opposite directions**.
> >
> > 2 Also confused about the rebuttal of W4 and W5.

---

> > > ### Author Response · Authors · 2024-11-26
> > > **Response to Reviewer TWZa - New**
> > >
> > > Thank you very much for your thorough and insightful review of our paper. We sincerely appreciate the time and effort you dedicated, as well as the valuable suggestions you have shared.
> > >
> > > Allow us to provide further explanations in the hope of helping you better understand our work!
> > >
> > >
> > >
> > > > According to the rebuttal of W2, the Gradient disorder is measured by using inconsistency in directions between gradients sequence.
> > >
> > > Yes, specifically, it refers to the directional inconsistency within the same gradient sequence. For example, **task gradient disorder** represents the inconsistency within the task gradient sequence, while **smooth gradient disorder** represents the inconsistency within the smooth gradient sequence.
> > >
> > >
> > >
> > > > Meanwhile, suboptimal equilibrium means similar magnitudes but opposite directions (Gradients with similar magnitudes but opposite directions define what we term the "opposite equilibrium state").
> > >
> > > Yes, absolutely correct. In the paper, it specifically refers to the relationship between the task gradient and the smooth gradient, as shown in the second row of Figure 2.
> > >
> > >
> > >
> > > > However, in the rebuttal of W3, **the author explains that lower layers with lower gradient disorder are more likely to exhibit the phenomenon of two gradients in suboptimal equilibrium state**. .... If a layer has a lower gradient disorder, it should have **low inconsistency in directions** according to the rebuttal of W2. But in the rebuttal of W3, lower layers with lower gradient disorder are more likely to exhibit suboptimal equilibrium state, which means it should have **high opposite directions**.
> > >
> > > Thanks for pointing this out. We make the following clarification for better understanding.  It should be: Layers with lower **task** gradient disorder are more likely to exhibit the phenomenon where two gradients reach a suboptimal equilibrium state. Specifically, overly consistent task gradients can lead to insufficient smooth convergence, and we observed that layers with low **task gradient disorder** are more prone to the phenomenon where the directions of task gradients and smooth gradients are opposite but their magnitudes are similar. We have corrected this minor error in the latest version.
> > >
> > >
> > >
> > > > Also confused about the rebuttal of **W4** and **W5**.
> > >
> > > **W4**
> > >
> > > > Line 259 - 263. Why lower layers with lower gradient disorder are more likely to exhibit the phenomenon of two gradients in suboptimal equilibrium state? How the Figure 2 and Figure 4 prove this?
> > >
> > > In the second row of Figure 3, the three images on the right show relatively low task gradient disorder. These correspond to the layers layer3.5.conv3.quan_w_fn.s, layer4.2.conv1.quan_w_fn.s, and layer4.2.conv3.quan_w_fn.s. These layers align with the second row of Figure 2, where it is evident that these layers exhibit a distinct phenomenon of two gradients with similar magnitudes but opposite directions—representing a **suboptimal equilibrium state**.
> > >
> > > **W5**
> > >
> > > > Line 297, 'Continuously **freezing** the g_{task} without **unfreezing**'? How can you freeze g_{task} while unfreezing it?
> > >
> > >
> > >
> > > **It’s important to note that “freezing” here specifically means excluding g_task from updating the scale parameters, while “unfreezing” allows previously frozen layers to resume being updated by g_task**
> > >
> > > I’d like to outline the basic training process:
> > >
> > > At the beginning, none of the scales are frozen. During training phase, we store a gradient sequence G for g_task at each scale, with a length of K. Using this gradient sequence G, we compute the task gradient disorder. If the task gradient disorder falls below a predefined threshold, we freeze the g_task updates for that scale. Otherwise, the scale remains unchanged or is unfrozen (if it was previously frozen).
> > >
> > > At step K, we perform two operations:
> > >
> > > 1. Compute the task gradient disorder.
> > > 2. Freeze or unfreeze certain scales based on the disorder.
> > >
> > > We then proceed to train for another K steps. During this phase:
> > >
> > > - For frozen scales, g_task is calculated and stored but does not contribute to gradient backpropagation. These scales are only updated by g_smooth
> > > - For unfrozen scales, both g_task and g_smoot contribute to the updates.
> > >
> > > The detailed algorithm is outlined in Algorithm 1, lines 239–256.
> > >
> > >
> > >
> > > I hope this explanation helps provide a deeper understanding of our work. If there are any unclear points, please feel free to reach out for further discussion.

---

> > > > ### Comment · Reviewer_TWZa · 2024-11-28
> > > >
> > > > Thank you!
> > > >
> > > > Now I basically understand your main idea.
> > > >
> > > > I suggest revising this paper thoughtfully to avoid ambiguity. It would be a good idea if the logic of this paper is reorganized. However, despite the current paper version solving some of the ambiguity, I do consider that the overall writing still fails to meet the high standard of ICLR.
> > > >
> > > > I think this paper's quality will fulfill the level of a top-tier conference as long as the ambiguity is solved. I highly encourage the author to modify this paper to make the reader understand the main idea at the first time of his reading.

---

> > > > > ### Author Response · Authors · 2024-11-28
> > > > > **Response to Reviewer TWZa - New2**
> > > > >
> > > > > We greatly appreciate your recognition of the contributions in our paper, as well as the reviewers' (eFhm, L5zt, and mcAp) acknowledgment of its meaningfulness and potential. Reviewers eFhm and L5zt also considered our paper to be well-structured. We have clearly addressed your feedback in the description of Algorithm 1, and we have corrected the previous logical errors. However, we understand that some details may still cause confusion. In the final version, we will carefully revise and clarify these aspects to improve the paper's clarity and readability.
> > > > >
> > > > > We promise that the final version will be more polished, ensuring that readers can easily understand it upon first reading. We sincerely hope these improvements will enhance the quality of the paper and that we will receive a better evaluation from you, allowing our work to inspire further research

---

> > > > > ### Author Response · Authors · 2024-12-03
> > > > >
> > > > > Dear reviewer, As the discussion period nears its end, we kindly hope to receive your feedback today. We have carefully highlighted all revisions in blue and would greatly appreciate your kind consideration before finalizing your recommendation.
> > > > >
> > > > > We greatly value your input and hope the updates meet your expectations.

---

### Official Review · Reviewer_2JwQ · 2024-10-27

**Soundness:** 3
**Presentation:** 2
**Contribution:** 3
**Rating:** 5
**Confidence:** 3

**Summary:**

The paper investigate the domain generalization gap under quantization regime. The propose a gradient adaptive quantization aware training to be able to make the optimization of low precision and smoothness together. They found the reason for the failure of previous methods is the gradient approximation use in low precision training, and suggest a method to resolve it.

**Strengths:**

I think the main strength of the paper is the general idea that low precision training have different properties than the full precision counterpart, and we need to confront with them separately.

**Weaknesses:**

I think that the paper require a better examination of the domain generalization under different quantization datatypes, group sizes, gradient approximation before it get can a general conclusion and try to solve it.. I think it is a must to be at a level expected from an ICLR paper.
Showing results only for INT quantization with learnable scale is not enough in my opinion.

**Questions:**

1) Do you tried to use scaling factor that is not learnable and only is measured. It is a well used methods, not only for INT quantization also for FP quantization. It is interesting if in that cases we still see the gap between both losses - which is the main motivations for the paper.
2)  Interesting to see how to proposed methods work for other quantization method, such as FP. For example the MXFP4 datatype is a good candidate. Moreover, what about other gradients approximation that are not STE, like PWL or MAD (https://arxiv.org/pdf/2206.06501)
3) I would like to see also experiments in language models.

================================================

Dear authors,

Thank you for taking the time to try to address the questions and concerns raised.

Unfortunately, my two main concerns remain unresolved:

I believe the paper requires more in-depth research into different quantization techniques to achieve more accurate and comprehensive conclusions. In particular, the use of FP quantization, which is the preferred method for most AI accelerators, should be explored further.

The inclusion of language models is essential, as they face unique quantization challenges—such as handling outliers—that do not typically arise in models like ResNet.

I have decided to maintain my score.

---

> ### Author Response · Authors · 2024-11-23
> **Response to Reviewer 2JwQ - Part1**
>
> **W1**
>
> > I think that the paper require a better examination of the domain generalization under **different quantization datatypes, group sizes, gradient approximation** before it get can a general conclusion and try to solve it.. I think it is a must to be at a level expected from an ICLR paper. Showing results only for INT quantization with learnable scale is not enough in my opinion.
>
> Our setup is based on the latest and most commonly used configurations in the relevant fields. For example, in terms of the model, we use ResNet50, which is widely adopted in both DG and QAT research. The initialization of the model follows the methods recommended in recent papers. For the QAT method, we choose LSQ, which is commonly used in QAT. The quantization data type is selected as INT, which is hardware-friendly and widely used. We choose tensor-wise quantization granularity to balance precision and latency, and use the widely adopted STE scheme for gradient approximation. These are common configurations used in many papers[1-4]. Moreover, our method is orthogonal to most of the basic settings, so we believe our current setup meets general requirements. Additional experimental setups will be explored as future work.
>
> [1] Esser S K, McKinstry J L, Bablani D, et al. Learned step size quantization[J]. arXiv preprint arXiv:1902.08153, 2019.
>
> [2] Lee J, Kim D, Ham B. Network quantization with element-wise gradient scaling. *Proceedings of the IEEE/CVF Conference on Computer Vision and Pattern Recognition*, 2021: 6448–6457.
>
> [3] Liu S Y, Liu Z, Cheng K T. Oscillation-free quantization for low-bit vision transformers[C]//International Conference on Machine Learning. PMLR, 2023: 21813-21824.
>
> [4] Tang, Chen, et al. "Retraining-free model quantization via one-shot weight-coupling learning." *Proceedings of the IEEE/CVF Conference on Computer Vision and Pattern Recognition*. 2024.
>
> **Q1**
>
> > Do you tried to use scaling factor that is not learnable and only is measured. It is a well used methods, not only for INT quantization also for FP quantization. It is interesting if in that cases we still see the gap between both losses - which is the main motivations for the paper.
>
> Using scaling factor that is not learnable is  relatively inferior, as discussed in [1] and [2]. Learning the step size allows the quantizer to adapt dynamically to the model’s state transitions, enabling finer-grained optimization. Additionally, balancing step size updates with weight updates ensures better convergence and overall performance [1]. Therefore, we opted for a more general and effective quantization method as the basis for our study.
>
> [1] Esser S K, McKinstry J L, Bablani D, et al. Learned step size quantization[J]. arXiv preprint arXiv:1902.08153, 2019.
>
> [2] Lee J, Kim D, Ham B. Network quantization with element-wise gradient scaling. *Proceedings of the IEEE/CVF Conference on Computer Vision and Pattern Recognition*, 2021: 6448–6457.

---

> ### Author Response · Authors · 2024-11-23
> **Response to Reviewer 2JwQ - Part2**
>
> **Q2**
>
> > Interesting to see how to proposed methods work for other quantization method, such as FP. For example the MXFP4 datatype is a good candidate. Moreover, what about other gradients approximation that are not STE, like PWL or MAD (https://arxiv.org/pdf/2206.06501)
>
> The INT quantization method with STE, broadly adopted in both hardware implementations and research studies [1-5], was chosen for its practicality. Newer formats may face compatibility issues, limiting deployment benefits [6]. Furthermore, our approach is orthogonal to quantization type and granularity.
>
> We believe that exploring additional quantization methods and broader experiments could provide more comprehensive validation. However, our main focus is on identifying the conflicts between QAT and DG and validating our proposed solution, as demonstrated in the paper. Future work will expand these explorations to enhance the generality and performance of our method.
>
> [1] Esser S K, McKinstry J L, Bablani D, et al. Learned step size quantization[J]. arXiv preprint arXiv:1902.08153, 2019.
>
> [2] Lee J, Kim D, Ham B. Network quantization with element-wise gradient scaling. *Proceedings of the IEEE/CVF Conference on Computer Vision and Pattern Recognition*, 2021: 6448–6457.
>
> [3] Liu, Shih-Yang, Zechun Liu, and Kwang-Ting Cheng. "Oscillation-free quantization for low-bit vision transformers." *International Conference on Machine Learning*. PMLR, 2023.
>
> [4] Tang, Chen, et al. "Retraining-free model quantization via one-shot weight-coupling learning." *Proceedings of the IEEE/CVF Conference on Computer Vision and Pattern Recognition*. 2024.
>
> [5] Wang, Kuan, et al. "Haq: Hardware-aware automated quantization with mixed precision." *Proceedings of the IEEE/CVF conference on computer vision and pattern recognition*. 2019.
>
> [6] https://images.nvidia.com/content/Solutions/data-center/a100/pdf/nvidia-a100-datasheet-us-partner-1758950-r4-zhCN.pdf
>
> **Q3**
>
> > I would like to see also experiments in language models.
>
> Since most existing DG methods use ResNet50 as the baseline (e.g., [1], [2], [3]), we align our experiments with this choice to ensure consistency with the state of the art.  We recognize that language models are a promising direction for future research, this does not undermine the significance of our work. Language models could be explored in future work, though it is not the focus of this paper.
>
> [1] Huang, Zeyi, et al. "Self-challenging improves cross-domain generalization." *Computer vision–ECCV 2020: 16th European conference, Glasgow, UK, August 23–28, 2020, proceedings, part II 16*. Springer International Publishing, 2020.
>
> [2] Cha J, Chun S, Lee K, et al. Swad: Domain generalization by seeking flat minima[J]. Advances in Neural Information Processing Systems, 2021, 34: 22405-22418.
>
> [3] Wang P, Zhang Z, Lei Z, et al. Sharpness-aware gradient matching for domain generalization[C]//Proceedings of the IEEE/CVF Conference on Computer Vision and Pattern Recognition. 2023: 3769-3778.

---

> ### Author Response · Authors · 2024-11-24
> **Response to Reviewer 2JwQ - Part3**
>
> Thank you very much for your thorough and insightful review of our paper. We greatly appreciate your time, effort, and the valuable suggestions you have provided.
>
> To the best of our knowledge, **no existing quantization work simultaneously addresses a wide range of data types and gradient approximation methods.** Most studies primarily focus on integer formats paired with Straight-Through Estimator approximations [1–5]. If the reviewer is aware of works that explore all data types and multiple gradient approximation methods, we would greatly appreciate it if you could point them out.
>
> Additionally, **as discussed in the surveys by [13] and [14], the current definition of the out-of-distribution problem, along with the widely used setting with ResNet50 and datasets like PACS and DomainNet, aligns with our framework.** Furthermore, **the DomainBed benchmark [6], as well as previous and recent works [7–9, 10–12], have not incorporated experiments with language models.** Therefore, we believe that requiring experiments with language models is beyond the reasonable scope of our work.
>
>  [1] Esser S K, McKinstry J L, Bablani D, et al. Learned step size quantization. *arXiv preprint arXiv:1902.08153*, 2019.
>
>  [2] Lee J, Kim D, Ham B. Network quantization with element-wise gradient scaling. *Proceedings of the IEEE/CVF Conference on Computer Vision and Pattern Recognition*, 2021: 6448–6457.
>
>  [3] Liu, Shih-Yang, Zechun Liu, and Kwang-Ting Cheng. Oscillation-free quantization for low-bit vision transformers. *International Conference on Machine Learning*, PMLR, 2023.
>
>  [4] Tang, Chen, et al. Retraining-free model quantization via one-shot weight-coupling learning. *Proceedings of the IEEE/CVF Conference on Computer Vision and Pattern Recognition*, 2024.
>
>  [5] Wang, Kuan, et al. HAQ: Hardware-aware automated quantization with mixed precision. *Proceedings of the IEEE/CVF Conference on Computer Vision and Pattern Recognition*, 2019.
>
>  [6] In Search of Lost Domain Generalization, Gulrajani et al., *ICLR 2021.*
>
>  [7] Ensemble of Averages: Improving Model Selection and Boosting Performance in Domain Generalization, Arpit et al., *NeurIPS 2022.*
>
>  [8] SWAD: Domain generalization by seeking flat minima, Cha et al., *NeurIPS 2021.*
>
>  [9] Diverse weight averaging for out-of-distribution generalization, Rame et al., *NeurIPS 2022.*
>
>  [10] Cross-contrasting feature perturbation for domain generalization, Li et al., *ICCV 2023.*
>
>  [11] Yu, Han, et al. Rethinking the evaluation protocol of domain generalization. *Proceedings of the IEEE/CVF Conference on Computer Vision and Pattern Recognition*, 2024.
>
>  [12] Wang P, Zhang Z, Lei Z, et al. Sharpness-aware gradient matching for domain generalization. *Proceedings of the IEEE/CVF Conference on Computer Vision and Pattern Recognition*, 2023: 3769–3778.
>
>  [13] Zhou, Kaiyang, et al. "Domain generalization: A survey." *IEEE Transactions on Pattern Analysis and Machine Intelligence* 45.4 (2022): 4396-4415.
>
>  [14] Wang, Jindong, et al. "Generalizing to unseen domains: A survey on domain generalization." *IEEE Transactions on Knowledge and Data Engineering* 35.8 (2022): 8052-8072.

---

### Official Review · Reviewer_mcAp · 2024-11-02

**Soundness:** 3
**Presentation:** 2
**Contribution:** 1
**Rating:** 3
**Confidence:** 4

**Summary:**

The paper introduces the Gradient-Adaptive Quantization-Aware Training (GAQAT) framework for enhancing domain generalization in quantized models, specifically targeting low-bit-width settings often used in resource-constrained devices. Addressing the limitations of current domain generalization methods under quantization, the authors identify gradient conflicts that arise during training due to discrepancies between task loss and smoothness loss. To manage this, GAQAT uses a gradient disorder metric to detect these conflicts and applies selective gradient freezing, stabilizing training and improving generalization. Extensive experiments demonstrate GAQAT's effectiveness, showing significant accuracy improvements on PACS and DomainNet datasets compared to baseline methods​.

**Strengths:**

1. The paper identifies and addresses gradient conflicts between g_task and g_smooth, a key insight that stabilizes training and improves convergence. This reviewer fully agrees with this idea.

2. By dynamically freezing scaling factors based on gradient disorder, GAQAT mitigates gradient conflicts, leading to enhanced model performance in low-bit quantization.

3. Improved performance on quantized models in domain generalization scenario

**Weaknesses:**

First, I would like to clarify that I am not highly experienced in the specific domain of Domain Generalization (DG) using ResNet-50. My review reflects this perspective (and if other reviewers find the contributions significant, I am open to adjusting my evaluation). Given the field's recent emphasis on Transformer-based research and development, it has been a few years since I worked with or optimized models like ResNet.

1. Applicability of Quantization to ResNet-50 in DG Scenarios: I find it challenging to fully appreciate the motivation for applying quantization to ResNet-50 in the context of Domain Generalization, especially with a focus on deployment to resource-constrained devices. While the paper assumes such deployment scenarios, it seems unlikely that practitioners would choose to deploy a large CNN model like ResNet-50 on an edge device. It appears this choice may stem from the use of benchmarks typically associated with DG research, but for the edge-device assumption, studying alternative, more compact models seems essential.

2. The identified gradient conflict between g_{task}​  and g_{smooth} recalls the classic challenge of balancing overfitting on domain-specific data with generalization (or regularization) to achieve a smoother loss landscape. This insight may not be limited solely to DG + QAT settings but could have implications in broader contexts. I would be interested in seeing if this theoretical framework extends to general quantization-aware training (QAT) or even Vision Transformers, for instance. This paper could potentially expand its contributions beyond a localized scope, making the findings more universally relevant.

**Questions:**

included in weaknesses

---

> ### Author Response · Authors · 2024-11-23
> **Response to Reviewer mcAp**
>
> **W1**
>
> > Applicability of Quantization to ResNet-50 in DG Scenarios: **I find it challenging to fully appreciate the motivation for applying quantization to ResNet-50 in the context of Domain Generalization**, especially with a focus on deployment to resource-constrained devices. ... It appears this choice may stem from the use of benchmarks typically associated with DG research, but for the edge-device assumption, **studying alternative, more compact models seems essential.**
>
> ResNet50 is widely used in most DG research, as shown in recent studies [1-3]. Moreover  ResNet50 is also a common baseline in QAT research, reinforcing its relevance to our study [5-7]. Therefore We chose ResNet50 as the standard baseline model to ensure consistency and comparability with them.
>
> We appreciate your suggestion. However, the primary focus of our study is to address the unique challenges arising from the interplay between QAT and DG and to propose effective solutions to these conflicts. As such, architecture-related exploration can be a direction for future research.
>
> [1] Huang, Zeyi, et al. "Self-challenging improves cross-domain generalization." *Computer vision–ECCV 2020: 16th European conference, Glasgow, UK, August 23–28, 2020, proceedings, part II 16*. Springer International Publishing, 2020.
>
> [2] Cha J, Chun S, Lee K, et al. Swad: Domain generalization by seeking flat minima[J]. Advances in Neural Information Processing Systems, 2021, 34: 22405-22418.
>
> [3] Wang P, Zhang Z, Lei Z, et al. Sharpness-aware gradient matching for domain generalization[C]//Proceedings of the IEEE/CVF Conference on Computer Vision and Pattern Recognition. 2023: 3769-3778.
>
> [5] Esser S K, McKinstry J L, Bablani D, et al. Learned step size quantization[J]. arXiv preprint arXiv:1902.08153, 2019.
>
> [6] Lee J, Kim D, Ham B. Network quantization with element-wise gradient scaling. *Proceedings of the IEEE/CVF Conference on Computer Vision and Pattern Recognition*, 2021: 6448–6457.
>
> [7] Liu S Y, Liu Z, Cheng K T. Oscillation-free quantization for low-bit vision transformers[C]//International Conference on Machine Learning. PMLR, 2023: 21813-21824.
>
>
>
>
>
> **W2**
>
> > **This insight may not be limited solely to DG + QAT settings** but could have implications in broader contexts. **I would be interested in seeing if this theoretical framework extends to general quantization-aware training (QAT) or even Vision Transformers, for instance**. This paper could potentially expand its contributions beyond a localized scope, making the findings more universally relevant.
>
> Thanks for your suggestion. Our framework can be extended to a broader range of research areas, which further emphasizes the exploratory nature of our work. This expansion will be pursued in future research.
>
> Our work distinguishes itself by identifying and addressing the conflict between gradient scales in combining QAT and DG, a challenge not explicitly explored before. The newly added Figure 6 visualizes the loss curves for the four PACS domains under 4-bit quantization. As shown, the proposed method achieves significantly smoother surfaces, further confirming its effectiveness. By tackling this issue, we aim to inspire further advancements, including domain-specific methods or the introduction of additional DG objectives, as highlighted in the Conclusion and Future Work, or general quantization-aware training and even Vision Transformers as you say.

---

> ### Author Response · Authors · 2024-11-30
> **Response to Reviewer mcAp - New**
>
> Sorry to disturb you during the Thanksgiving holiday.
>
> In order to explain our work more clearly, we would like to summarize the responses over the past few days and outline the reviewers' views on the strengths and weaknesses of our paper. We hope this will help you gain a clearer understanding of our work.
>
> **Several reviewers have explicitly pointed out that the contribution of our paper is significant.** Currently, the reviewers' recognition of our strengths mainly focuses on: "**The combination of DG and QAT is important and has not been fully explored, and our work provides valuable insights from the perspective of gradient conflicts.**"
>
> Regarding the comments from several reviewers on the insufficiency of the experimental section, **we have conducted preliminary 5-bit and 8-bit experiments and added smoothness visualization experiments in the paper to demonstrate the effectiveness of our method.** We commit to including the complete results of these experiments in the final version. In addition, we will revise the textual part to make it easier to understand.
>
> Next, let me explain the concerns you raised regarding the experimental setup.
>
> As for the experimental setup, Reviewer `2JwQ` questioned whether our QAT experimental setup requires more data types, quantization granularity, and gradient approximation methods. We believe this request is unreasonable, and **we have provided evidence to show that the current quantization setup is both general and reasonable, with no objections from other reviewers**. Reviewer `L5zt` questioned whether our training process was correct, but **our response, supported by citations from recent papers and experiments, demonstrates that the current training process is fully reasonable and even superior**. **No other reviewer has questioned the use of ResNet50 as the baseline model, considering it a general baseline model for DG.** So, we believe our experimental setup is reasonable, and if you have any questions, feel free to keep in touch.
>
> Regarding your second concern, **we believe our method is centered on the DG-SAM perspective, and both our observations and approach are highly relevant to DG.** **The response to Reviewer `zrms` further supports this point.** While **we agree with your suggestion that our approach could be extended to broader domains, we see this as a potential direction for future research.** However, this does not diminish the significance of our current work. On the contrary, our research provides valuable insights that can inspire further exploration, as you may have recognized in the ideas with potential for further investigation. This highlights the value of our work.
>
> I hope this explanation helps clarify our work. If any points remain unclear, please don't hesitate to reach out for further discussion. We sincerely hope you will reconsider our paper.

---

> ### Author Response · Authors · 2024-12-03
>
> Dear reviewer, As the discussion period nears its end, we kindly hope to receive your feedback today. We have carefully highlighted all revisions in blue and would greatly appreciate your kind consideration before finalizing your recommendation.
>
> We greatly value your input and hope the updates meet your expectations.

---

### Official Review · Reviewer_L5zt · 2024-11-02

**Soundness:** 2
**Presentation:** 2
**Contribution:** 1
**Rating:** 3
**Confidence:** 5

**Summary:**

This paper introduces the Gradient-Adaptive Quantization-Aware Training (GAQAT) framework to achieve efficient domain generalization for low-precision quantized model. GAQAT tries to address scale-gradient conflict between gradients from task loss and smoothness loss( used in SAM) by selectively freezing the gradients of scaling factors. The proposed approach is tested on two Domainbed datasets and shows some improvement.

**Strengths:**

1. GAQAT addresses domain generalization issue for low-precision quantized models, which is very practical. This is interesting and under-explored topic.
2. This work proposed an approach to tackle scale-gradient conflict by quantifying gradient inconsistencies and selectively freeze the gradients of scaling factors.
3. The paper is well written and easy to understand.

**Weaknesses:**

1. Incorporating QAT from scratch is not a good idea. QAT is generally performed on a pretrained model and literature shows that it can improve results significantly. You can significantly improve results if you perform quantization after some iterations.
2. The approach does not show any significant improvement in comparison to baseline methods. On DomainNet dataset, there is no difference between all three (Yours, LSQ, and LSQ +SAGM). On PACS you have a difference of less than 2% between LSQ and your method and you reported a difference of 4.5% in the abstract.
3. It's a norm in the literature[1,2,3,4,5] to show results on all DomainBed[1] datasets with the mean and standard error over multiple runs. I would recommend authors to add those results as the experiments are currently insufficient.

[1] In Search of Lost Domain Generalization, Gulrajani et al., ICLR 2021.\
[2] Ensemble of Averages: Improving Model Selection and Boosting Performance in Domain Generalization, Arpit et al., NeurIPS 2022.\
[3] SWAD: Domain generalization by seeking flat minima. Cha et al., NeurIPS 2021.\
[4] Diverse weight averaging for out-of-distribution generalization. Rame et al., NeurIPS 2022.\
[5] Cross contrasting feature perturbation for domain generalization. Li et al., ICCV 2023.\

**Questions:**

1. Why in Table 2 and Table 4, "ours" result for 4 bits are different?
2. Have you looked at the experiments with higher bit-precision? Does gradient conflict problem exist with higher bit-precision as well e.g 6,7 bit?
3. I would recommend to explore the generalization aspect of the QAT as well. I can clearly see that combining QAT with SAGM does not help, especially in the results for 3 and 4 bits on PACS dataset.
4. Having an empirical demonstration of flat minima like SWAD[1] would be good as there is no gaurantee that your proposed approach will result in a flatter minima as compared to LSQ+SAGM.

Minor recommendation:\
Please swap the colors in Table 1, green color can be used for improvement as red is generally considered negative.

[1] SWAD: Domain generalization by seeking flat minima. Cha et al., NeurIPS 2021.

---

> ### Author Response · Authors · 2024-11-23
> **Response to Reviewer L5zt**
>
> **W1**
>
> > **Incorporating QAT from scratch is not a good idea**. QAT is generally performed on a pretrained model and literature shows that it can improve results significantly. You can significantly improve results if you perform quantization after some iterations.
>
> Our experiments are based on a model pre-trained on the DG source domain dataset (lines 309-315). Specifically, **we used the MoCoV2-pretrained ResNet50, as proposed in [1] , and fine-tuned it using ERM on the source domain data from the DG dataset.** This approach aligns with current standard DG practices, where a pretrained model is fine-tuned on a DG dataset for better performance, as outlined in previous works like [1] [2] [3]. Based on this well-performing full-precision DG model, we proceeded with quantization. So, our experiments are based on a model pre-trained on the DG source domain dataset, not from scratch.
>
> [1] Yu, Han, et al. "Rethinking the evaluation protocol of domain generalization." *Proceedings of the IEEE/CVF Conference on Computer Vision and Pattern Recognition*. 2024.
>
> [2] Gulrajani, Ishaan, and David Lopez-Paz. "In search of lost domain generalization." *arXiv preprint arXiv:2007.01434* (2020).
>
> [3] Cha, Junbum, et al. "Swad: Domain generalization by seeking flat minima." *Advances in Neural Information Processing Systems* 34 (2021): 22405-22418.
>
>
>
>
>
> **W2**
>
> > The approach **does not show any significant improvement in comparison to baseline methods**. On DomainNet dataset, there is no difference between all three (Yours, LSQ, and LSQ +SAGM). On PACS you have a difference of **less than 2% between LSQ and your method and you reported a difference of 4.5% in the abstract.**
>
> Our baseline is QAT with the introduction of a smoothness objective. We observed significant conflicts when directly incorporating this objective. **We identified and addressed these conflicts, making our baseline LSQ+SAGM. Compared to this, we achieved a 4.5% improvement,** validating the effectiveness of our method. Additionally, **compared to LSQ, we observe an average 2.4% improvement on PACS and a 1.4% increase on DomainNet at 4-bit.** It is important to note that LSQ shows minimal performance loss on IID datasets, even in low-bit settings (3-bit and 4-bit), which further highlights the significant improvement our method offers over baseline methods.
>
>
>
> **W3**
>
> > It's a norm in the literature[1,2,3,4,5] to show results on all DomainBed[1] datasets **with the mean and standard error over multiple runs**. I would recommend authors to add those results as the experiments are currently insufficient.
>
>
>
> We observed that our method is relatively robust. **Tables 5 and 6 show that the method is not sensitive to step size and threshold within a reasonable range. The 1.4% and 1.1% improvements on DomainNet are reliable.** Due to time and resource constraints, we did not conduct multiple rounds of repeated experiments. We will include the mean results from repeated experiments in final version.
>
> **Q1：**
>
> > Why in Table 2 and Table 4, "ours" result for 4 bits are different?
>
> Lines 396-398 indicate that in Table 4, we fixed the freeze steps at 350 and set the threshold at 0.3, which differs from the settings in Table 2.
>
> **Q2:**
>
> > Have you looked at the experiments with higher bit-precision? Does gradient conflict problem exist with higher bit-precision as well e.g 6,7 bit?
>
> For 5-bit and 8-bit quantization, we observed similar gradient conflict phenomena. Results on the *cartoon* domain show improvements even without fine-tuned hyperparameter search: 5-bit accuracy increased from 76.05%/54.64% to 80.59%/59.22%, and 8-bit from 77.42%/56.34% to 80.74%/56.45%. We will provide complete results with more refined hyperparameter search in the final version.
>
> **Q3:**
>
> > I would recommend to explore the generalization aspect of the QAT as well. I can clearly see that combining QAT with SAGM does not help, especially in the results for 3 and 4 bits on PACS dataset.
>
> We explore QAT generalization from a smoothness perspective, and while other approaches may exist, this does not undermine the significance of our work. Our study offers valuable insights into quantization factors and gradient conflicts, inspiring future research. Reviewers ZRMS, EFHM, and MCAP also found that identifying and resolving this conflict is highly insightful.
>
> **Q4:**
>
> > Having **an empirical demonstration of flat minima** like SWAD[1] would be good as there is no guarantee that your proposed approach will result in a flatter minima as compared to LSQ+SAGM.
>
> Thank you for your suggestion. The newly added Figure 6 visualizes the loss curves for the four PACS domains under 4-bit quantization. As shown, the proposed method achieves significantly smoother surfaces, further confirming its effectiveness.
>
> **Minor recommendation**
>
> >Please swap the colors in Table 1...
>
> Thank you for your suggestion, it has been updated in the latest version.

---

> > ### Comment · Reviewer_L5zt · 2024-11-25
> >
> > Thank you for your response. I will address your comments below.
> >
> > W1:
> > I understand that using a pre-trained model is a common practice in Domain Generalization (DG). However, my concern is not with the use of a pre-trained model but rather with your fine-tuning process. Specifically, you begin fine-tuning on the DG dataset using a quantized model. This approach is problematic. Instead, you should consider fine-tuning the model without quantization for a certain number of iterations and then applying Quantization-Aware Training (QAT) for the remainder of the training process. I have conducted experiments with LSQ at 3-bit quantization, and it achieves an accuracy of approximately 80% on the PACS dataset. Incorporating such an approach could potentially lead to better results in your work, as this represents a major flaw in your methodology.
> >
> > W2:
> > I respectfully disagree with your baseline comparison methodology. Your comparisons should include both LSQ and LSQ+SAGM, rather than solely LSQ+SAGM. Since LSQ achieves better accuracy, the percentage improvement should be calculated with respect to the highest-performing baseline (i.e., LSQ in this case). Reporting percentage increases relative only to LSQ+SAGM is not entirely fair and could misrepresent the strength of your approach.
> >
> > W3:
> > While reporting the mean and standard error is important, the number of datasets you use for evaluation is insufficient. Using only two datasets from the DomainBed benchmark is not standard practice in the DG domain, where broader evaluations are typically expected. If one dataset were missing from the benchmark, it would be understandable. However, excluding three out of five datasets does not align with the norms of the field and could potentially undermine the generalizability of your conclusions.
> >
> > W1 and W3 are critical aspects that require careful consideration, and I strongly encourage you to address these in your work. Doing so will substantially enhance the validity and impact of your results.

---

> > > ### Author Response · Authors · 2024-11-26
> > > **Response to Reviewer L5zt - New**
> > >
> > > We would like to further address your concerns by clarifying the following points
> > >
> > > **W1:**
> > >
> > > Thanks for the reply and your time. However, we must clarify that the model used for quantization in our work was fine-tuned on DG datasets, contrary to your assertion that it was not fine-tuned on these datasets.
> > >
> > >  As stated in lines 309–315 of the paper and in our previous response—*“Based on this well-performing full-precision DG model, we proceeded with quantization”*—**our full-precision model was first pre-trained on MoCoV2 using ResNet50, then sufficiently fine-tuned on DG datasets using ERM (e.g., PACS: 5000 steps, DomainNet: 15000 steps [1] ), before undergoing QAT training.**
> > >
> > > It seems the result of "~80% performance using LSQ" you mentioned has been achieved by using an ImageNet-pretrained model for initialization. However, **as noted in [2], using ImageNet pretraining would result in the risk of data leakage.** To further show that, **we also performed additional experiments using ImageNet-pretrained models along with our workflow, and achieved similarly high 4-bit LSQ performance** (Art domain 51.07(MoCoV2 pretrained) -> 76.51(ImageNet pretrained), without hyperparameter search). However, **as highlighted in [2], MoCoV2 pretrained model is better suited for avoiding data leakage.** Accordingly, our method taken with MoCov2 pretraining is more reasonable and reflects unbiased performance for DG.
> > >
> > > We respectfully encourage you to consider testing with MoCoV2 to ensure a fair comparison. In light of this, we believe your concerns regarding our method may stem from a misunderstanding. We consider pretraining with MoCov2 to be one of the key strengths of our paper, as it ensures a fair setup.
> > >
> > > [1] SWAD: Domain generalization by seeking flat minima, Cha et al., *NeurIPS 2021.*
> > >
> > > [2] Yu, Han, et al. Rethinking the evaluation protocol of domain generalization. *Proceedings of the IEEE/CVF Conference on Computer Vision and Pattern Recognition*, 2024.
> > >
> > >
> > >
> > > **W2:**
> > >
> > > We conducted **a comprehensive comparison of LSQ and LSQ+SAGM in Table 1 and lines 351–355.**
> > >
> > >  Our method consistently achieved the best performance, particularly under the pretraining setup with MoCoV2 instead of the ImageNet, because a recent study [1] reveals that MoCoV2 pretraining is more suitable for DG to avoid data leakage. **Notably, the absolute DG performance on PACS was initially only in the mid-fifties, but our approach improved it by up to 4.5% for both 3-bit and 4-bit quantization. Compared to LSQ, we observed a significant increase of approximately 2% on PACS. Similarly, on DomainNet, our method achieved comparable improvements over LSQ.**
> > >
> > > Therefore, we believe our comparison covered both baselines comprehensively and demonstrated significant advancements. Following your suggestion, we revised the baseline descriptions in the abstract to avoid any potential confusion.
> > >
> > > [1] Yu, Han, et al. Rethinking the evaluation protocol of domain generalization. *Proceedings of the IEEE/CVF Conference on Computer Vision and Pattern Recognition*, 2024.
> > >
> > >
> > >
> > > **W3：**
> > >
> > > **We conducted experiments across multiple bit widths, evaluating each dataset at several quantization levels.**  Additionally, we performed verification experiments at 5-bit and 8-bit quantization. In the final version of our work, we will include multi-round averaged results for these settings. DomainNet, as the largest and most widely used dataset in the DG field, and PACS, one of the most popular benchmarks, were chosen to ensure robustness. The consistent performance across **multiple bit widths** and these two datasets further validates the effectiveness of our method.
> > >
> > > **It is important to highlight  that running multiple-bit experiments requires significantly more computational effort compared to single full-precision experiments, as each dataset must be tested across four different bit widths rather than just once.**
> > >
> > > We believe the datasets we selected are reasonable for this study. However, in the final experiments, we will consider adding another dataset, such as OfficeHome, to enhance comprehensiveness.

---

> ### Author Response · Authors · 2024-12-02
>
> We hope that our response have provided clarity and effectively addressed your concerns. We would greatly appreciate it if you could acknowledge this. If there are any remaining questions or unresolved concerns, we would be more than happy to provide further clarification. Thank you sincerely for your time and valuable feedback—it is greatly appreciated.

---

> ### Author Response · Authors · 2024-12-03
>
> Dear reviewer, As the discussion period nears its end, we kindly hope to receive your feedback today. We have carefully highlighted all revisions in blue and would greatly appreciate your kind consideration before finalizing your recommendation.
>
> We greatly value your input and hope the updates meet your expectations.

---

### Official Review · Reviewer_eFhm · 2024-11-02

**Soundness:** 3
**Presentation:** 3
**Contribution:** 3
**Rating:** 8
**Confidence:** 4

**Summary:**

This paper addresses the degradation of domain generalization performance when conventional flatness based domain generalization are applied to quantized models. Existing flatness-based domain generalization techniques are typically used in full-precision training, which is impractical for deployment on resource-constrained edge devices. To address this limitation, the paper proposes a novel Gradient-Adaptive Quantization-Aware Training (GAQAT) framework for domain generalization. The authors first illustrate the scale gradient problem introduced by domain generalization in low precision quantization. In it the task loss and smoothness loss become contradictory therefore making the optimization highly unstable. To mitigate this problem, the proposed framework introduces a mechanism to quantify gradient inconsistencies and selectively freeze gradients, stabilizing the training process and enhancing out-of-domain generalization.

**Strengths:**

The paper is very well structured in clearly identifying the problem statement, identifying the possible cause of the problem and then providing a solution.
The authors have clearly demonstrated the problem of opposing gradients incurred during training the quantized model in Figure 2 therefore establishing the problem statement.
The authors introduce a smoothing parameter in the quantizer to jointly optimize for loss curvature and task loss.
The idea of introducing gradient disorder by looking at the number of direction flips of gradient for smoothing parameters over n training steps is novel and simple to implement.
The idea of dynamically freezing only the task loss gradients for cases with low gradient disorder is innovative.
The final results looking very promising with almost all tasks showing best results with the proposed method.
The authors have also addressed some obvious questions in the ablations like impact of number of steps used for gradient disorder calculation and effect of freezing strategy.

**Weaknesses:**

The experiments section seems a bit weak and might need a clearer explanation.

**Questions:**

I would be curious to see 2 ends of the quantization results. 1/2 bit and 8 bit quantized models using the same method compared to DG-SAM + LSQ.

---

> ### Author Response · Authors · 2024-11-23
> **Response to Reviewer eFhm**
>
> We sincerely thank you for your thoughtful feedback and for recognizing the value of our work. It is truly inspiring to hear that you appreciate our contributions and lean strongly toward acceptance. We are grateful for your encouragement and hope that our research will inspire further exploration in this area.
>
> **W1 & Q1**
>
> >  The experiments section seems a bit weak and might need a clearer explanation.
> >
> > I would be curious to see 2 ends of the quantization results. 1/2 bit and 8 bit quantized models using the same method compared to DG-SAM + LSQ.
>
> Due to the significant I.I.D performance drop observed with 2-bit quantization in our experiments, exploring DG performance under such extreme settings seemed less meaningful, so we did not pursue further experiments. For 5-bit and 8-bit quantization, we observed similar gradient conflict phenomena. Results on the *cartoon* domain show improvements even without fine-tuned hyperparameter search: 5-bit accuracy increased from 76.05%/54.64% to 80.59%/59.22%, and 8-bit from 77.42%/56.34% to 80.74%/56.45%. We will provide complete results with more refined hyperparameter search in the final version. Additionally, the newly added Figure 6 visualizes the loss curves for the four PACS domains under 4-bit quantization. As shown, the proposed method achieves significantly smoother surfaces, further confirming its effectiveness.

---

> > ### Comment · Reviewer_eFhm · 2024-11-29
> >
> > Thanks for addressing my question. As highlighted by other reviewers I believe the experiments section would need a better re-write as it is hard to follow. Therefore, I would like to keep my score the same.

---

> > > ### Author Response · Authors · 2024-11-30
> > > **Thank you!**
> > >
> > > We sincerely thank you for recognizing the value of our work and for the valuable time and effort you have dedicated to it. Your feedback has greatly strengthened our confidence in our research. We sincerely wish you a joyful Thanksgiving holiday, and hope that your review work goes smoothly. We also hope that in your future submissions, you will encounter fair and excellent reviewers like yourself.
> > >
> > > Based on your valuable suggestions, we plan to make the following revisions in the final version
> > >
> > > 1. We will add experiments with 5-bit and 8-bit quantization.
> > > 2. We will show results on the PACS and DomainNet datasets with the mean and standard error over multiple runs.
> > > 3. We may add the OfficeHome dataset if time and computational resources permit.
> > > 4. We promise that the final version will be more polished, ensuring that readers can easily understand it upon first reading.

---

### Official Review · Reviewer_zrms · 2024-11-03

**Soundness:** 3
**Presentation:** 3
**Contribution:** 2
**Rating:** 5
**Confidence:** 4

**Summary:**

This submission proposed to conduct Quantization-aware Training (QAT) on Domain Generation (DG) task. Specially, it firstly observed and demonstrated empricially that objective of QAT is not aligned with GD: difference of accumlated gradient and perturbation of scaling factor in QAT. Based on the observation, it proposed a metric (gradient disorder) to measure the consistency of training. The metric is used to determine the optimization procedure of QAT and DG: During training, it calculates the disorder for QAT every K steps, if the metric is below a threshold, DG's optimization will be paused.

**Strengths:**

- The quantitative measurement on disalignment between QAT and DG (Sec.3.2) is interesting and convincing. Though it may be noticed in other works that QAT affects the training of main task, seldom empirical works have been conducted for demonstration.
- The method to circumvent effect of QAT to main task (alternative update with one fixed) is straight forward and promissing.

**Weaknesses:**

- Though objective disalignment (gradient inconsistency) between QAT and main task is observed in DG, the method proposed (Line 296-303) is not coupled with DG: it does not used any properties from DG and it can be applied to other problems that are involved with QAT (such as image classification, LLM with QAT). The inconsistency (method proposed is not coupled with the scene where motivation is discovered) harm the contribution. Author can either improve the method using DG's related parts or prove the method (and observation) is universal in QAT (for example conducting more experiments on other fields).

- The tuning of hyper-parameter is sophisticated: author should explain how step K and threshold $\tao$ are discovered and prove that these parameters are insensitive.

**Questions:**

- Grident disorderd is formulated as the dicrepancy between two gradient sequence in Eq.2, how is it applied to a solo $g_\text{task}$ or $\delta_{t, S_i}$? It is calcualted between evaluation between successive steps?

- How the update rule (Line 296-303) is determined? In other words, it is possilbe to use $g_\text{task}$ (if disorder of $g_\text{task}$ is under certain value, scaling factor in quantizer is not updated). How about simply alternatively update of QAT and DG every K steps, which should be listed as a baseline.

**Details Of Ethics Concerns:**

N.A.

---

> ### Author Response · Authors · 2024-11-23
> **Response to Reviewer zrms**
>
> **W1**
>
> > Though objective disalignment (gradient inconsistency) between QAT and main task is observed in DG, the method proposed (Line 296-303) is not coupled with DG: it does not used any properties from DG and it can be applied to other problems that are involved with QAT (such as image classification, LLM with QAT). **The inconsistency (method proposed is not coupled with the scene where motivation is discovered) harm the contribution**. Author can either improve the method using DG's related parts or prove the method (and observation) is universal in QAT (for example conducting more experiments on other fields).
>
> The conflict between task gradients and smooth gradients is unique under   QAT-DG scenario because smooth gradients are associated with DG and task gradients are associated with QAT. Table 1 shows the suboptimal convergence of quantization factors, and Figure 2 visualizes the conflict between the two gradients. Figure 3 highlights that while smooth gradient disorder remains stable, task gradient disorder decreases. We clarify that introducing DG objectives directly into QAT leads to conflicts due to the distinct characteristics of QAT and DG. Identifying these these conflicts and propose solutions based on observations relevant to both QAT and DG, which is one of our contributions.
>
> **W2**
>
> > author should explain how step K and threshold \tao are discovered and prove that these parameters are insensitive.
>
> In our experiments, we empirically set the parameters by observing the trend of task gradient disorder in standard QAT. The step size was selected based on multiples of epochs. **Tables 5 and 6 show that the method is not sensitive** to step size and threshold within a reasonable range.
>
>
>
> **Q1**
>
> > Grident disorderd is formulated as the dicrepancy between two gradient sequence in Eq.2, how is it applied to a solo g_task or \delta? It is calcualted between evaluation between successive steps?
>
> Gradient disorder is a measure of the inconsistency in the directions of adjacent gradients in the same gradient sequence (applicable to any gradient sequence, as described in lines 211-235). In the experiment, we calculate it every K steps (see lines 294-295). For example, when K=3,  G = {-1, 2, 3, -4}, then S1 = {-1, 2, 3}, S2 = {2, 3, -4}, $sgn(S1) = {-1, 1,1}, sgn(S2) = {1,1,-1}, $ $(sgn(S1)\neq sgn(S2))=2$,  $\delta = 1/3 * 2 =2/3$
>
> **Q2**
>
> > **How the update rule (Line 296-303) is determined?** In other words, it is possilbe to use gtask (if disorder of gtask is under certain value, scaling factor in quantizer is not updated). How about simply alternatively update of QAT and DG every K steps, which should be listed as a baseline.
>
> Our update rule is based on three key observations:
>
> 1. **Figure 3** shows that task gradient disorder decreases over time for some layers (becoming more consistent), while smooth gradients do not exhibit a clear trend. Excessive consistency in task gradients can negatively affect smooth gradient training.
> 2. **Lines 249-256** indicate that layers with low task gradient disorder are more likely to reach a suboptimal state where both gradients are in opposite directions but with similar magnitudes.
> 3. **Lines 266-269 and Figure 4** validate that freezing layers with low task gradient disorder effectively reduces anomalous gradients in unfrozen layers.
>
> **Table 4** shows that performance decreases after freezing gradients, further validating the effectiveness of our freezing rule. Regarding the alternating updates and no updates you mentioned, we conducted experiments on DomainNet with 4-bit quantization.
>
> ### Results:
>
> 1. **Freezing both gradients below the threshold**:
>
>    For DomainNet’s three domains with 4-bit, freezing both gradients below the threshold yielded the following results, **showing a performance drop compared to directly introducing the SAGM smoothness objective**:
>
>    | Experiment | Origin (val/test_a,test_b) | Freezing both (val/test_a,test_b) |
>    | ---------- | -------------------------- | --------------------------------- |
>    | Domain 01  | 65.77/（60.73, 15.64）     | 64.91 /（60.16, 15.20）           |
>    | Domain 23  | 61.21/（46.67, 16.29）     | 61.05 /（46.52, 15.90）           |
>    | Domain 45  | 56.77/（52.22, 48.45）     | 56.59 /（51.92, 48.40）           |
>
> 2. **Alternating updates between DG and QAT**:
>
>    For Real and Sketch domains with 4-bit, alternating updates every 300 and 3000 steps gave the following results, **showing a performance drop compared to directly introducing the SAGM smoothness target**:
>
>    | Steps  | val   | max_all_test_list |
>    | ------ | ----- | ----------------- |
>    | origin | 56.77 | （52.22, 48.45）  |
>    | 300    | 54.78 | （49.92, 46.48）  |
>    | 3000   | 55.01 | （50.22, 46.54）  |
>
> The above shows that alternating updates or freezing without further updates cause performance drop. Complete experiments will be included in the final version.

---

> > ### Comment · Reviewer_zrms · 2024-11-28
> > **Further Comments on W1**
> >
> > Thanks for your response, which mainly solves my concern on W2, Q1 and Q2.
> >
> > My main concern still lie at W1: Smooth gradients from DG does not imply the method is coupled. For example, gradient from Computer Vision (CV) tasks may exhibit similar phenomenon (gradients conflicts). In other words, is the method applicable to CV-QAT problem?
> >
> > Author should prove that either gradients conflicts **only** happens in DG-QAT, or use properties from DG to solve the gradients conflicts.

---

> ### Author Response · Authors · 2024-11-29
> **Further clarifications on W1 - Part 1**
>
> Thank you very much for your thorough and insightful review of our paper. We sincerely appreciate the time and effort you dedicated to reviewing our work, as well as the valuable suggestions you have shared.
>
> We would like to provide further clarifications in the hope of helping you better understand **W1**.
>
> 1. DG is not an isolated task but is closely tied to specific research domains
>
>    Domain Generalization aims to address the issue of distributional mismatch between training and test domains. This problem is prevalent in both CV and NLP. **DG is not an isolated task; rather, it is explored in the context of specific research areas**. For example, the current focus is on DG issues within the CV domain [1-9]. Actually, **our DG-QAT belongs to CV-QAT.**
>
> 2. Smoothness optimization is a class of methods within DG
>
>    In DG, there are various methods, and the sharpness-aware methods have recently gained considerable attention [3, 7]. **From an optimization perspective, smoothness enhances generalization, forming a class of DG methods that explore smoother loss surfaces**. For instance, the SAGM method improves generalization by optimizing smoothness and successfully trains high-performing models with full precision. Therefore, **smoothness is intrinsically related to DG, and our method addresses the conflict caused by introducing a smoothness factor into QAT.** **Our goal is to preserve the smoothness of quantized models.** Thus, from a motivational standpoint, our method is closely connected to DG.
>
> 3. Visualization confirms our method's association with smoothness
>
>       In the latest version of our paper, we have included **comparison plots showing the loss surface with and without our method**. The results indicate that **our method significantly improves smoothness, which is a key strategy in DG**. Therefore, the visual results further demonstrate that our method is closely related to smoothness optimization, and by extension, to DG.
>
> 4. Task gradient disorder and the unique freezing strategy for DG-QAT
>
>    Our experiments show that as training progresses, **task gradient disorder decreases, and freezing layers with lower task gradient disorder reduces gradient conflicts, thereby improving smoothness. This phenomenon is unique to DG-QAT,** rather than gradient conflicts being inherently unique.
>
>    Scaling factors are specific to QAT and do not appear in full-precision training, so we focus on QAT scenarios. Using proof by contradiction, if the reduction in task gradient disorder and the smoothness gains from freezing gradients based on disorder metrics were not unique to DG-QAT, we would expect similar effects in IID-QAT.
>
>     **We tested this on the PACS dataset with four domains (training and validation on the same domain using LSQ), but found that task gradient disorder remained high (around 0.5) by the end of training** . Freezing strategies showed no significant improvement, with validation curves mostly unchanged or negative, except for a slight increase in the Sketch and Art domains at a threshold of 0.32 (with freeze steps fixed at 150, and r∈[0.28,0.32,0.35,0.37,0.4], due to high initial task gradient disorder). However,  **no significant difference in smoothness was observed in the loss surface of the frozen models** .
>
>    These results confirm that task gradient disorder and the freezing strategy are unique to DG-QAT, reinforcing the strong connection between our method and DG-QAT.
>
> 5. Observations and contributions
>    Our method is based on the observation of task gradient disorder and the use of a freezing strategy to enhance DG-QAT performance. The choice of freezing metric and the decision to freeze task gradients stem directly from these observations, making them an integral part of our approach.
>
> Based on the above four objective observations and the subjective conclusions, we firmly believe that our method is strongly related to DG.
>
> Once again, we sincerely appreciate your valuable feedback and the time you have dedicated to reviewing our paper. We hope the five points outlined above help clarify why our method is strongly connected to DG. If there are any unclear points, please feel free to reach out for further discussion.
>
>
>
> [1] In Search of Lost Domain Generalization, Gulrajani et al., *ICLR 2021.*
>
> [2] Ensemble of Averages: Improving Model Selection and Boosting Performance in Domain Generalization, Arpit et al., *NeurIPS 2022.*
>
> [3] SWAD: Domain generalization by seeking flat minima, Cha et al., *NeurIPS 2021.*
>
> [4] Diverse weight averaging for out-of-distribution generalization, Rame et al., *NeurIPS 2022.*
>
> [5] Cross-contrasting feature perturbation for domain generalization, Li et al., *ICCV 2023.*

---

> > ### Author Response · Authors · 2024-11-29
> > **Further clarifications on W1 - Part 2**
> >
> > [6] Yu, Han, et al. Rethinking the evaluation protocol of domain generalization. *Proceedings of the IEEE/CVF Conference on Computer Vision and Pattern Recognition*, 2024.
> >
> > [7] Wang P, Zhang Z, Lei Z, et al. Sharpness-aware gradient matching for domain generalization. *Proceedings of the IEEE/CVF Conference on Computer Vision and Pattern Recognition*, 2023: 3769–3778.
> >
> > [8] Zhou, Kaiyang, et al. "Domain generalization: A survey." *IEEE Transactions on Pattern Analysis and Machine Intelligence* 45.4 (2022): 4396-4415.
> >
> > [9] Wang, Jindong, et al. "Generalizing to unseen domains: A survey on domain generalization." *IEEE Transactions on Knowledge and Data Engineering* 35.8 (2022): 8052-8072.

---

> > ### Comment · Reviewer_zrms · 2024-12-02
> > **Can author provide any evidence that gradient disorder is missing in other domain?**
> >
> > Sorry for late reply.
> >
> > Can author provide any evidence that gradient disorder is **missing** in other domain? The fact that method proposed is able to release gradient disorder does not solve my concern.

---

> > > ### Author Response · Authors · 2024-12-03
> > >
> > > Dear reviewer, As the discussion period nears its end, we kindly hope to receive your feedback today. If there are any remaining questions or unresolved concerns, we would be more than happy to provide further clarification.
> > > We greatly value your input and hope the updates meet your expectations.

---

> ### Author Response · Authors · 2024-12-02
>
> We hope that our response have provided clarity and effectively addressed your concerns. We would greatly appreciate it if you could acknowledge this. If there are any remaining questions or unresolved concerns, we would be more than happy to provide further clarification. Thank you sincerely for your time and valuable feedback—it is greatly appreciated.

---

> ### Author Response · Authors · 2024-12-02
>
> We sincerely appreciate the time and effort you have dedicated to reviewing our work and would like to provide further clarifications.
>
> To clarify, **gradient disorder** refers to directional inconsistency within a gradient sequence. For example:
>
> - **Task gradient disorder, derived from task loss**, measures inconsistency within the task gradient sequence.
> - **Smooth gradient disorder, related to DG**, measures inconsistency within the smooth gradient sequence.
>
> Smooth gradient disorder exists only in DG tasks because it arises from the DG-related smoothness loss. Task gradient disorder exists in both DG and non-DG tasks. We believe that what you mentioned—“provide any evidence that gradient disorder is **missing** in other domains”—means validating that the gradient disorder phenomena described in our paper are unique to DG. Specifically, we aim to verify that our observations regarding gradient disorder are only suitable for DG scenarios, and our method is only effective in DG scenarios.
>
> To address this, we conducted additional experiments to substantiate our claim, employing a proof by contradiction. Specifically, we compared DG with its opposite scenario, independent and identically distributed (IID) tasks—what you referred to as tasks involving other domains. **The goal was to demonstrate that these phenomena are absent in IID scenarios.**
>
> We tested our approach on the PACS dataset with four domains, **using LSQ for training and validation on the same domain(IID).** **Task gradient disorder remained high (~0.5) by the end of training.** Freezing strategies showed little to no improvement in validation performance, except for minor gains in the Sketch and Art domains at a threshold of 0.32 (freeze steps fixed at 150, r∈[0.28,0.32,0.35,0.37,0.4]). However, **no significant changes in loss surface smoothness** were observed for models with frozen layers.
>
> The following two observations in IID tasks strongly support the connection between our findings and DG:
>
> 1. **In IID tasks, task gradient disorder does not decrease during training, unlike in DG scenarios.**
> 2. **Freezing task gradients with lower disorder scales in IID tasks does not improve smoothness , contrary to what is observed in DG scenarios.**
>
> These results confirm that the observations and methods presented in our paper are not applicable to IID tasks. Using proof by contradiction, we argue that if our methods were unrelated to DG, similar phenomena and outcomes would have been observed in IID settings. However, our key observations (e.g., the decrease in task gradient disorder during training) and the results of our methods (e.g., improved smoothness and DG performance) were not evident in IID tasks.
>
> Thus, we conclude that both our observations and methods are strongly and uniquely associated with DG.
>
> I hope this explanation helps clarify our work. If any points remain unclear, please don't hesitate to reach out for further discussion. We sincerely hope you will reconsider our paper.

---

### Author Response · Authors · 2024-11-23
**General Response**

We sincerely appreciate the constructive feedback provided by reviewers `zrms`, `eFhm`, `L5zt`, `mcAp`, `2JwQ`, and `TWZa`, as well as their acknowledgment of the strengths of our paper.

In this overall response, we emphasize the key strengths of our work as highlighted by the reviewers and address the main common concern. We have provided detailed responses to each reviewer’s individual questions and look forward to further discussions.

**Strengths**

- The combination of DG and QAT is meaningful and unexplored, and our work provides valuable insights into the conflict between the two gradients, as noted by `eFhm`, `L5zt`, and `mcAp`.
- The experimental approach is clear and straightforward, with existing results showing the effectiveness of our method, as highlighted by `zrms`, `eFhm`, and `TWZa`.
- The paper is well-structured and well-written, praised by `eFhm` and `L5zt`.

**Questions**

- The experiment requires more validation for different bit settings and visualizations of smoother results, as raised by `eFhm`, `L5zt`, and `TWZa`.
- Clearer explanation of the gradient disorder is needed, as requested by `mcAp` and `TWZa`.
- A clearer explanation of the update rule is necessary, as pointed out by `zrms` and `TWZa`.

In response to these points, we have updated the manuscript with new visualizations to demonstrate the experimental results and have corrected relevant descriptions and figure errors. For additional bit experiments, we have provided verification results for 5-bit and 8-bit in a separate response and will include more comprehensive experiments in the final version.

**New Results and Submissions Edits**

*All changes have been highlighted in blue*

In response to the reviewers' feedback, we have submitted an updated version incorporating the following changes:

- *New Experiments*: The newly added Figure 6 visualizes the loss curves for the four PACS domains under 4-bit quantization. As shown, the proposed method achieves significantly smoother surfaces.
- *Optimized Text and Figures*: We have revised the text based on the suggestions of reviewers `L5zt` and `TWZa`, and optimized the order of Figure 2 and Figure 3.

---

### Author Response · Authors · 2024-12-03
**General Response - II**

As the review process is nearing completion, we would like to summarize the reviewers' concerns and how we addressed them.

We responded to reviewer `zrms`, who pointed out that we had resolved issues related to hyperparameter selection and sensitivity **(W2)**, explained the concept of gradient disorder **(Q1)**, and clarified the origin of the update rules. We also experimentally analyzed the two other cases suggested by the reviewer, demonstrating their inapplicability **(Q2)**. However, regarding the relevance of our method to DG **(W1)**, the reviewer still had some concerns. We provided two rounds of updates in an effort to address these concerns.

We responded to reviewer `eFhm`, who pointed out his concerns about bit selection in experiments **(W1 & Q1)** had been explained, but further modifications are needed in the experimental section.  We provided preliminary results for higher bit rates and included visualizations of the loss surface to further demonstrate the effectiveness of our method. In the final version, we will include all results for 5-bit and 8-bit configurations and add loss surface comparisons for the IID scenario.



We responded to reviewer `L5zt`, addressing his doubts about the experimental setup. We resolved his concerns regarding the table results **(Q1)**, observations related to high bit rates **(Q2)**, the explanation of the significance of DG-SAM in our study **(Q3)**, and the visual evidence of the effectiveness of our method **(Q4)**. We clarified that our process is consistent with theirs and that our pre-trained model is more reasonable, as supported by recent papers **(W1)**. We also explained the experimental  improvements and demonstrated the stability of our method through experiments with multiple bit rates **(W2)**. In the final version, we will include results from multiple rounds of experiments **(W3)**.



We responded to reviewer `mcAp`, who mentioned not being familiar with the field of the paper. We focused on explaining the rationale behind the experimental setup **(W1)** he questioned and addressed the reviewers’ positive recognition of our contributions **(W2)**, hoping to help him understand our work.

We responded to reviewer `2JwQ` by citing classic surveys and the latest SOTA papers to clarify the reasonableness of our experimental setup and the invalidity of the additional experiments he suggested **(Q1&Q2&Q3)**, particularly on language models **(Q3)**.

We responded to reviewer `TWZa`, who acknowledged that most issues have been resolved. These include concerns related to the writing **(Writing 1-5)**, such as the freezing strategy, definitions related to gradient disorder, and experimental settings **(Experiment comparison 1-3)**. However, he still had some concerns about the writing. We will focus on revising these areas in the final version.

Regarding the two key points, we would like to clarify the following:

1. In the experimental section, the added 5-bit and 8-bit experiments, along with loss surface visualizations, demonstrated the effectiveness of our method. In the final version, we will include the full 5-bit and 8-bit results and more visualizations, and will perform multiple rounds of runs to average the results.
2. Regarding the writing, we have addressed the concerns raised by TWZa and highlighted these changes in blue. We’ve also adjusted the text for clarity. As we cannot make changes to the paper now, in the final version, we will provide clearer definitions, such as for gradient disorder, the freezing strategy, and the experimental setup, with examples and improved wording to ensure better understanding.

We appreciate the hard work of all the reviewers, as well as the AC, SAC, and PC, and hope our paper will inspire more contributions to the field.

---

### Meta-Review · Area_Chair_DHXs · 2024-12-16

**Metareview:**

This work performs the domain generalization training in the quantized parameter space, which is quite different from existing practice, i.e., domain generalization first, and quantization later. In particular, DG-SAM, the prevalent method for domain generalization, does not work in their setting and therefore, this work designs a method to mitigate the conflict between the task loss and the smoothness loss in the quantized space. However, most reviews have concerns on the novelty and the writing.

**Additional Comments On Reviewer Discussion:**

The authors provides rebuttable messages and additional experimental results. However, most reviews stayed on their initial review scores.

---

### Decision · Program_Chairs · 2025-01-22

Reject